# Bandit Social Learning under Myopic Behavior

**Kiarash Banihashem**
University of Maryland, College Park
kiarash@umd.edu

**MohammadTaghi Hajiaghayi**
University of Maryland, College Park
hajiagha@umd.edu

**Suho Shin**
University of Maryland, College Park
suhoshin@umd.edu

**Aleksandrs Slivkins**
Microsoft Research NYC
slivkins@microsoft.com

## Abstract

We study social learning dynamics motivated by reviews on online platforms. The agents collectively follow a simple multi-armed bandit protocol, but each agent acts myopically, without regards to exploration. We allow a wide range of myopic behaviors that are consistent with (parameterized) confidence intervals for the arms' expected rewards. We derive stark exploration failures for any such behavior, and provide matching positive results. As a special case, we obtain the first general results on failure of the greedy algorithm in bandits, thus providing a theoretical foundation for why bandit algorithms should explore.[1]

## 1 Introduction

Reviews and ratings are pervasive in many online platforms. A customer consults reviews/ratings, then chooses a product and then (often) leaves feedback, which is aggregated by the platform and served to future customers. Collectively, customers face a tradeoff between *exploration* and *exploitation*, *i.e.,* between acquiring new information while making potentially suboptimal decisions and making optimal decisions using information currently available. However, individual customers tend to act myopically and favor exploitation, without regards to exploration for the sake of the others. On a high level, we ask **whether/how the myopic behavior interferes with efficient exploration**. We are particularly interested in *learning failures* when only a few agents choose an optimal action.

We distill this issue down to its purest form. We posit that the customers make one decision each and do not observe any personalized payoff-relevant information prior to their decision, whether public or private. In particular, the customers believe they are similar to one another. They have only two alternative products/experiences to choose from, a.k.a., *arms*, and no way to infer anything about one arm from the other. The platform provides each customer with full history on the previous agents.[2]

Concretely, we posit *Bandit Social Learning* (BSL): a variant of social learning in which the customers (henceforth, *agents*) arrive sequentially and follow a simple multi-armed bandit protocol. Each agent observes full history, chooses an arm, and receives a reward: a Bernoulli random draw whose mean is arm-specific and unknown. Initial knowledge (a dataset with some samples of each arm) may be available to all agents. When all agents are governed by a centralized algorithm, this setting is known as *stochastic bandits*, a standard and well-understood variant of multi-armed bandits.

---

[1]Early versions of our results on the greedy algorithm (Corollary 3.6 and Theorem 6.1) have been available in a book chapter by A. Slivkins [54, Ch. 11]. The authors acknowledge Mark Sellke for proving Theorem 6.1 and suggesting a proof plan for a version of Corollary 3.6. The authors are grateful to Mark Sellke and Chara Podimata for brief collaborations (with A. Slivkins) in the initial stages of this project.

[2]In practice, online platforms provide summaries such as the average score and the number of samples.

37th Conference on Neural Information Processing Systems (NeurIPS 2023).

| Mean rewards | Beliefs | Behavior | Result |
|---|---|---|---|
| fixed | "frequentist" confidence intervals | $\eta$-confident | Thm. 3.1 (main), Thm. 3.9 (small $N_0$). |
| | | unbiased/Greedy | Cor. 3.6 |
| | | $\eta_t$-pessimistic | Thm. 3.10 |
| | Bayesian (independent) | Bayesian-unbiased | Thm. 5.1(a) |
| | | $\eta$-Bayesian-confident | Thm. 5.1(b) |
| Bayesian (correlated) | Bayesian (and correct) | Bayesian-unbiased | Thm. 6.1 |

Table 1: Our negative results: learning failures.

We allow a wide range of myopic behaviors that are consistent with observations. Given an arm, consider the confidence interval for its mean reward, parameterized by $\eta \geq 0$: the sample average plus/minus the "confidence term", $\sqrt{\eta/\#\text{samples}}$. [3] An *$\eta$-confident* agent evaluates each arm to an *index*: some number within this arm's confidence interval (and otherwise arbitrary), and chooses an arm with a largest index. (Computational implementation of this process is irrelevant to our model.) Crucially, the $\eta$ and the agents' behavior are given and cannot be influenced by the platform.

This model subsumes the "unbiased" behavior, when the index equals the sample average, as well as various "behavioral biases" (see "related work" for citations). Most notably: "optimism" and "pessimism", when the index is, resp., larger or smaller than the sample average. (These can also be interpreted as, resp., risk seeking and risk aversion.) The model also allows for probabilistic decisions (via randomized indices), correlated behaviors (when samples from one arm affect the behavioral bias on another), and recency bias (when one favors more recent observations). Further, an agent may treat each arm differently, and different agents may exhibit different biases.

We target the regime when parameter $\eta$ is constant w.r.t. the number of agents $T$. *I.e.,* the agents' population is characterized by a constant $\eta$. We are interested in the asymptotic behavior when $T$ increases. An extreme version of our model, with $\eta \sim \log(T)$, is only considered for intuition and sanity checks. Interestingly, this version subsumes two well-known bandit algorithms: UCB1 [6] and Thompson Sampling [57, 52], which achieve optimal regret bounds. These algorithms can be seen as behaviors: resp., extreme optimism and *probability matching* [49, 59], a well-known randomized behavior. More "moderate" versions of these behaviors are consistent with $\eta$-confidence as defined above, and are subject to the learning failures described below.

**Our results.** We are interested in learning failures when all but a few agents choose the bad arm, and how the failure probability scales with the $\eta$ parameter. Our main result is that with $\eta$-confident agents, the failure probability is at least $e^{-O(\eta)}$ (see Section 3). Consequently, regret is at least $\Omega(T \cdot e^{-O(\eta)})$ for any given problem instance, in contrast with the $O(\log T)$ regret rate obtained by optimal bandit algorithms. Further, the $e^{-O(\eta)}$ scaling is the best possible: indeed, regret for optimistic agents is at most $O\left(T \cdot e^{-\Omega(\eta)} + \eta\right)$ for a given problem instance (Theorem 4.1). Note that the negative result deteriorates as $\eta$ increases, and becomes vacuous when $\eta \sim \log T$; the upper bound then essentially matches the optimal $O(\log T)$ regret of the UCB algorithm [6]).

We refine these results in several directions. First, if all agents are "unbiased", the failure probability scales as the difference in expected reward between the two arms (Corollary 3.6). Second, if all agents are pessimistic, then any level of pessimism, whether small or large or different across agents, leads to the similar failure probability as in the unbiased case (Theorem 3.10). Third, a small fraction of optimists goes a long way! That is, if all agents are $\eta$-confident and even a $q$-fraction of them are $\eta$-optimistic, then we obtain regret $O\left(T \cdot e^{-\Omega(\eta)} + \eta/q\right)$ regardless of the other agents. [4]

Our results extend to Bayesian agents who have independent priors across the arms and act according to their posteriors. Such agents are consistent with our main model of $\eta$-confident agents, and therefore are subject to the same negative results (Section 5). Further, we focus on Bayesian-unbiased agents and allow arbitrary *correlated* Bayesian beliefs (when the agents can make inferences about one arm from the observations on the other). We derive a general result on learning failures, assuming that the mean rewards are actually drawn according to the beliefs (Section 6).

---

[3] Then the confidence interval contains the (true) mean reward with probability at least $1 - 2e^{-2\eta}$.

[4] A similar result holds even the agents hold different levels of optimism, see Theorem 4.5.

| Mean rewards | Beliefs | Behavior | Result |
|---|---|---|---|
| fixed | "frequentist" | $\eta$-optimistic | Thm. 4.1 |
| | confidence | $\eta_t$-optimistic, $\eta_t \in [\eta, \eta_{\max}]$ | Thm. 4.4 |
| | intervals | small fraction of optimists | Thm. 4.5 |

Table 2: Our positive results: upper bounds on regret.

Our results are summarized in Tables 1 and 2.

**Implications for multi-armed bandits.** The negative results for unbiased agents can be seen as general results on the failure of the *greedy algorithm*: a bandit algorithm that always exploits. This is a theoretical foundation for why bandit algorithms should explore – and indeed why one should design them. We are not aware of any general results of this nature, whether published or known previously as "folklore", which is quite surprising given the enormous literature on multi-armed bandits. Therefore, we believe our results fill an important gap in the literature.

How surprising are these results? It has been folklore knowledge for several decades that the greedy algorithm is inefficient in some simple special cases, and folklore *belief* that this should hold much more generally. However, recent results reveal a more complex picture: the greedy algorithm fails under some strong assumptions, but works well under some other strong assumptions (see Related Work). Thus, it has arguably became less clear which assumptions would be needed for negative results and what would be the "shape" and probability of learning failures.

Further, our results on $\eta$-confident agents explain why UCB1 algorithm requires extreme optimism, why any algorithm based on narrow (constant-$\eta$) confidence intervals is doomed to fail, and also why "pessimism under uncertainty" is not a productive approach for exploration.

**Novelty and significance.** BSL was not well-understood previously even with unbiased agents, as discussed above, let alone for more permissive behavioral models. It was very unclear a priori how to analyze learning failures and how strong would be the guarantees, in terms of the generality of agents' behaviors, the failure events/probabilities, and the technical assumptions.

On a technical level, our proofs have very little to do with standard lower-bound analyses in bandits stemming from [43, 7]. These analysis apply any algorithm and prove "sublinear" lower bounds on regret, such as $\Omega(\log T)$ for a given problem instance and $\Omega(\sqrt{T})$ in the worst case. Their main technical tool is KL-divergence analysis showing that no algorithm can distinguish between a given tuple of "similar" problem instances. In contrast, we prove *linear* lower bounds on regret, our results apply to a particular family of behaviors/algorithms, and we never consider a tuple of similar problem instances. Instead, we use anti-concentration and martingale tools to argue that the best arm is never played (or played only a few times), with some probability. The result on correlated beliefs in Section 6 has a rather short but "conceptual" proof which we believe is well-suited for a textbook.

While our positive results in Section 4 are restricted to "optimistic" agents, we do not assert that such agents are necessarily typical. The primary point here is that our results on learning failures are essentially tight. That said, "optimism" is a well-documented behavioral bias (*e.g.,* see [50] and references therein). So, a small fraction of optimists (leveraged in Theorem 4.5) is not unrealistic.

Our proofs are more involved compared to the standard analysis of the UCB1 algorithm. This is because we cannot make the $\eta$ parameter as large as needed to ensure that the complements of certain "clean events" can be ignored. Instead, we need to define and analyze these "clean events" in a more careful way. These difficulties are compounded in Theorem 4.5, our most general result. As far as the statements are concerned, the basic result in Theorem 4.1 is perhaps what one would expect to hold, whereas the extensions in Theorem 4.4 and Theorem 4.5 are more surprising.

**Framing.** We target the scenario in social learning when both actions and rewards are observable in the future, and the agents do not receive any other payoff-relevant signals. As in much of algorithmic game theory, we discuss the influence of self-interested behavior on the overall welfare of the system. We consider how such behaviour can cause "learning failures", which is a typical framing in the literature on social learning. From the perspective of multi-armed bandits, we investigate the failures of the greedy algorithm, and more generally any algorithm that operates on narrow confidence intervals. We do not attempt to design new algorithms, as a version of UCB1 is proved optimal.

**Map of the paper.** Section 2 defines our model. Section 3 derives the learning failures. Section 4 provides positive results for optimistic agents. Section 5 and Section 6 handle agents with Bayesian beliefs. Most proofs are moved to the supplement.

**Related work.** A vast literature on social learning studies agents that learn over time in a shared environment. A prominent topic is learning failures such as ours. Models vary across several dimensions: *e.g.,* the information acquired/transmitted, the communication network, agents' life-span and decision rules, etc. All models from prior work are very different from ours. In the supplement, we separate our model from several most relevant ones: "sequential social learning" [27], "strategic experimentation" [33], networked myopic learners [8, 46], and misspecified beliefs [31, 15, 25, 44].

Positive results for the greedy bandit algorithm [37, 11, 51] focus on *contextual bandits*, an extension of stochastic bandits where a payoff-relevant signal (*context*) is available before each round. Equivalently, each agent in BSL receives such signal along with the history (incl. all previous signals). Very strong assumptions are needed: linearity of rewards and diversity of contexts. A similar result holds for BSL with *private* signals, under different (and also very strong) assumptions on structure and diversity [1]. In all this work, agents' diversity substitutes for exploration, and structural assumptions allow aggregation across agents. Moreover, the greedy algorithm obtains $o(T)$ regret in various scenarios with a very large number of near-optimal arms [13, 35], *e.g.,* in Bayesian bandits with $K \gg \sqrt{T}$ arms and independent uniform priors. We focus on a more basic model, with only two arms and no contexts, where all these channels are ruled out.

Learning failures for the greedy algorithm are derived for bandit problems with 1-dimensional action spaces under (strong) structural assumptions: *e.g.,* dynamic pricing with linear demands [30, 22] and dynamic control in a (generalized) linear model [42, 40]. In all these results, the failure probability is only proved positive, but not otherwise characterized.

*Incentivized exploration* takes a mechanism design perspective on BSL, whereby the platform strives to incentivize individual agents to explore for the sake of the common good. In most of this work, starting from [41, 19], the platform controls the information flow, *e.g.,* can withhold history and instead issue recommendations, and uses this information asymmetry to create incentives; see [55], [54, Ch. 11] for surveys.[5] In particular, [48, 34, 53] target stochastic bandits as the underlying learning problem, same as we do. In [34], the platform constructs a (very) particular communication network for the agents, and then the agents engage in BSL on this network.

Non-Bayesian models of behavior are prominent in social learning, starting from DeGroot [21]: agents use variants of statistical inference and/or naive rules-of-thumb to infer the state of the world. In particular, our model of $\eta$-confident agents is essentially a special case of "case-based decision theory" [26]. Well-documented behavioral biases allowed by our model include: optimism and pessimism (*e.g.,* [50] and [18, 12], resp., and references therein), risk aversion/risk seeking [36, 10], recency bias (*e.g.,* [24] and references therein), randomized decisions (with theory tracing back to Luce [47]), and *probability matching* more specifically [49, 59].

Our perspective of multi-armed bandits is very standard in machine learning theory: we consider asymptotic regret rates without time-discounting. The vast literature on regret-minimizing bandits is summarized in books [17, 54, 45]. Stochastic bandits is a standard, basic version with i.i.d. rewards and no auxiliary structure. Most relevant are the UCB1 algorithm [6], Thompson Sampling [57, 52] (particularly the "frequentist" analyses thereof [2, 4, 38]), and the lower-bound results [43, 7].

## 2   Our model and preliminaries

Our model, called **Bandit Social Learning**, is defined as follows. There are $T$ rounds, where $T \in \mathbb{N}$ is the time horizon, and two *arms* (*i.e.,* alternative actions). We use $[T]$ and $[2]$ to denote the set of rounds and arms, respectively.[6] In each round $t \in [T]$, a new agent arrives, observes history $\texttt{hist}_t$ (defined below), chooses an arm $a_t \in [2]$, receives reward $r_t \in [0, 1]$ for this arm, and leaves forever. When a given arm $a \in [2]$ is chosen, its reward is drawn independently from Bernoulli distribution

---

[5]Alternatively, the agents observe full history, but the platform uses payments to create incentives [23, 29, 20].
[6]Throughout, we denote $[n] = \{1, 2, \ldots, n\}$, for any $n \in \mathbb{N}$.

with mean $\mu_a \in [0,1]$. [7] The mean reward is fixed over time, but not known to the agents. Some initial data is available to all agents, namely $N_0 \geq 1$ samples of each arm $a \in [2]$. We denote them $r_{a,i}^0 \in [0,1]$, $i \in [N_0]$. The history in round $t$ consists of both the initial data and the data generated by the previous agents. Formally, it is a tuple of arm-reward pairs,

$$\texttt{hist}_t := \big( (a, r_{a,i}^0) : \; a \in [2], i \in [N_0]; \; (a_s, r_s) : \; s \in [t-1] \big).$$

We summarize the protocol for Bandit Social Learning as Protocol 1.

---

**Protocol 1:** Bandit Social Learning

---

Problem instance: two arms $a \in [2]$ with (fixed, but unknown) mean rewards $\mu_1, \mu_2 \in [0,1]$ ;
Initialization: $\texttt{hist} \leftarrow \{ N_0$ samples of each arm $\}$;
**for** *each round $t = 1, 2, \ldots, T$* **do**
     agent $t$ arrives, observes $\texttt{hist}$ and chooses an arm $a_t \in [2]$ ;
     reward $r_t \in [0,1]$ is drawn from Bernoulli distribution with mean $\mu_{a_t}$;
     new datapoint $(a_t, r_t)$ is added to $\texttt{hist}$

---

*Remark* 2.1. The initial data-points represent reports created outside our model, *e.g.,* by ghost shoppers or influencers, and available before the products enter the market. One could interpret them as a simple "frequentist" representation for the initial beliefs of the agents, with $N_0$ as the beliefs' "strength". We posit $N_0 \geq 1$ to ensure that the arms' average rewards are always well-defined.

If the agents were controlled by an algorithm, this protocol would correspond to *stochastic bandits* with two arms, the most basic version of multi-armed bandits. A standard performance measure in multi-armed bandits (and online machine learning more generally) is *regret*, defined as

$$\text{Regret}(T) := \mu^* \cdot T - \mathbb{E}\left[ \sum_{t \in [T]} \mu_{a_t} \right], \tag{2.1}$$

where $\mu^* = \max(\mu_1, \mu_2)$ is the maximal expected reward of an arm.

Each agent $t$ chooses its arm $a_t$ myopically. Each agent is endowed with some (possibly randomized) mapping from histories to arms, and chooses an arm accordingly. This mapping, called *behavioral type*, encapsulates how the agent resolves uncertainty on the rewards. More concretely, each agent maps the observed history $\texttt{hist}_t$ to an *index* $\texttt{Ind}_{a,t} \in \mathbb{R}$ for each arm $a \in [2]$, and chooses an arm with a largest index. The ties are broken independently and uniformly at random.

We allow for a range of myopic behaviors, whereby each index can take an arbitrary value in the (parameterized) confidence interval for the corresponding arm. Formally, fix arm $a \in [2]$ and round $t \in [T]$. Let $n_{a,t}$ denote the number of times this arm has been chosen in the history $\texttt{hist}_t$ (including the initial data), and let $\hat{\mu}_{a,t}$ denote the corresponding average reward. Given these samples, standard (frequentist, truncated) upper and lower confidence bounds for the arm's mean reward $\mu_a$ (UCB and LCB, for short) are defined as follows:

$$\texttt{UCB}_{a,t}^\eta := \min\left\{ 1, \hat{\mu}_{a,t} + \sqrt{\eta/n_{a,t}} \right\} \quad \text{and} \quad \texttt{LCB}_{a,t}^\eta := \max\left\{ 0, \hat{\mu}_{a,t} - \sqrt{\eta/n_{a,t}} \right\}, \tag{2.2}$$

where $\eta \geq 0$ is a parameter. The interval $\left[ \texttt{LCB}_{a,t}^\eta, \texttt{UCB}_{a,t}^\eta \right]$ will be referred to as $\eta$-*confidence interval*. Standard concentration inequalities imply that $\mu_a$ is contained in this interval with probability at least $1 - 2\,e^{-2\eta}$ (where the probability is over the random rewards, for any fixed value of $\mu_a$). We allow the index to take an arbitrary value in this interval:

$$\texttt{Ind}_{a,t} \in \left[ \texttt{LCB}_{a,t}^\eta, \texttt{UCB}_{a,t}^\eta \right], \quad \text{for each arm } a \in [2]. \tag{2.3}$$

We refer to such agents as $\eta$-*confident*; $\eta > 0$ will be a crucial parameter throughout.

**On special cases.** Our model accommodates a number of behavioural biases. Most notably: *unbiased agents*, who set $\texttt{Ind}_{a,t} = \hat{\mu}_{a,t}$, $\eta$-*optimistic agents*, who set $\texttt{Ind}_{a,t} = \texttt{UCB}_{a,t}^\eta$, and $\eta$-*pessimistic agents* who set $\texttt{Ind}_{a,t} = \texttt{LCB}_{a,t}^\eta$. Unbiased agents formally correspond to the *greedy algorithm*, whereas *extreme* optimism, i.e., $\eta$-optimism with $\eta \sim \log(T)$, corresponds to UCB1 algorithm [6].

---

[7]Our results on upper bounds (Section 4) and Bayesian learning failures (Section 6) allow each arm to have an arbitrary reward distribution on $[0,1]$. We omit further mention of this to simplify presentation.

Our model also allows a version of Thompson Sampling in which the posterior samples are truncated to the $\eta$-confidence interval.[8] More generally, we allow *Bayesian agents* that preprocess observations to a Bayesian posterior, and use the latter to define their indices. See the supplement for more details.

**Preliminaries.** When $\mu_1, \mu_2$ are fixed (not drawn from a prior), we posit $\mu_1 > \mu_2$, *i.e.,* arm 1 is the *good arm*, and arm 2 is the *bad arm*. Our guarantees depend on quantity $\Delta := \mu_1 - \mu_2$, called the *gap* (between the two arms). It is a very standard quantity for regret bounds in multi-armed bandits.

We use the big-O notation to hide constant factors. Specifically, $O(X)$ and $\Omega(X)$ mean, resp., "at most $c_0 \cdot X$" and "at least $c_0 \cdot X$" for some absolute constant $c_0 > 0$ that is not specified in the paper. When and if $c_0$ depends on some other absolute constant $c$ that we specify explicitly, we point this out in words and/or by writing, resp., $O_c(X)$ and $\Omega_c(X)$. As usual, $\Theta(X)$ is a shorthand for "both $O(X)$ and $\Omega(X)$", and writing $\Theta_c(X)$ emphasizes the dependence on $c$.

Algorithms UCB1 and Thompson Sampling achieve regret

$$\text{Regret}(T) \leq O(\,\min(\,1/\Delta, \sqrt{T}\,) \cdot \log T\,). \tag{2.4}$$

This regret rate is essentially optimal among all bandit algorithms: it is optimal up to constant factors for fixed $\Delta > 0$, and up to $O(\log T)$ factors for fixed $T$ (see "related work" for citations).

A key property of a reasonable bandit algorithm is that $\text{Regret}(T)/T \to 0$; this property is also called *no-regret*. Conversely, algorithms with $\text{Regret}(T) \geq \Omega(T)$ are considered very inefficient.

# 3 Learning failures

We are interested in learning failures when all but a few agents choose the bad arm. More precisely, we define the $n$-*sampling failure* as an event that all but at most $n$ agents choose the bad arm.

We make two technical assumptions:

mean rewards satisfy $c < \mu_2 < \mu_1 < 1 - c$ for some absolute constant $c \in (0, 1/2)$, $\qquad$ (3.1)

the number of initial samples satisfies $N_0 \geq 64\,\eta/c^2 + 1/c$. $\qquad$ (3.2)

The meaning of (3.1) is that it rules out degenerate behaviors when mean rewards are close to the known upper/lower bounds. The big-O notation hides the dependence on the absolute constant $c$, when and if explicitly stated so. Assumption (3.2) ensures that the $\eta$-confidence interval is a proper subset of $[0, 1]$ for all agents; we sidestep this assumption later in Theorem 3.9.

Our main result allows arbitrary $\eta$-confident agents and asserts that 0-sampling failures happen with probability at least $p_{\texttt{fail}} \sim e^{-O(\eta)}$. This is a stark failure when $\eta$ is a constant relative to $T$.

**Theorem 3.1** ($\eta$-confident agents)**.** *Suppose all agents are $\eta$-confident, for some fixed $\eta \geq 0$. Make assumptions (3.1) and (3.2). Then the $0$-sampling failure occurs with probability at least*[9]

$$p_{\texttt{fail}} = \Omega_c(\,\Delta + \sqrt{\eta/N_0}\,) \cdot e^{-O_c(\,\eta + N_0 \Delta^2\,)}, \quad \text{where} \quad \Delta = \mu_1 - \mu_2. \tag{3.3}$$

*Consequently,* $\text{Regret}(T) \geq \Delta \cdot p_{\texttt{fail}} \cdot T$.

We emphasize generality: the agents can exhibit any behaviors consistent with $\eta$-confidence, possibly different for different agents and different arms. From multi-armed bandit perspective, the theorem implies that bandit algorithms consistent with $\eta$-confidence cannot have regret sublinear in $T$.

The guarantee in Theorem 3.1 deteriorates as the parameter $\eta$ increases, and becomes vacuous when $\eta \sim \log(T)$. This makes sense, as this regime of $\eta$ is used in UCB1 algorithm.

*Discussion* 3.2. Assumption (3.2) is innocuous from the social learning perspective: essentially, the agents hold initial beliefs grounded in data and these beliefs are not completely uninformed. From the bandit perspective, this assumption is more substantive, as an algorithm can always choose to discard data. In any case, we remove this assumption in Theorem 3.9 below.

*Remark* 3.3. A weaker version of (3.2), namely $N_0 \geq \eta$, is necessary to guarantee an $n$-sampling failure for any $\eta$-confident agents. Indeed, suppose all agents are $\eta$-optimistic for arm 1 (the good arm), and $\eta$-pessimistic for arm 2 (the bad arm). If $N_0 < \eta$, then the index for arm 2 is 0 after the initial samples, whereas the index of arm 1 is always positive. Then all agents choose arm 1.

---

[8]For $\eta \sim \log T$, this coincides with Thompson Sampling with very high probability.

[9]Throughout the paper, we use the notation $O_c$ to hide the dependence on the absloute constant $c$.

Next, we spell out two corollaries which help elucidate the main result.

**Corollary 3.4.** *If the gap is sufficiently small, $\Delta < O\left(1/\sqrt{N_0}\right)$, then Theorem 3.1 holds with*

$$p_{\texttt{fail}} = \Omega_c(\ \Delta + \sqrt{\eta/N_0}\ ) \cdot e^{-O_c(\eta)}. \tag{3.4}$$

*Remark* 3.5. The assumption in Corollary 3.4 is quite mild in light of the fact that when $\Delta > \Omega\left(\sqrt{\log(T)/N_0}\right)$, the initial samples suffice to determine the best arm with high probability.

**Corollary 3.6.** *If all agents are unbiased, then Theorem 3.1 holds with $\eta = 0$ and*

$$p_{\texttt{fail}} = \Omega_c\left(\Delta\right) \cdot e^{-O_c\left(N_0\,\Delta^2\right)} \tag{3.5}$$
$$= \Omega_c\left(\Delta\right) \qquad \text{if } \Delta < O(\ 1/\sqrt{N_0}\ ).$$

*Remark* 3.7. A trivial failure result for unbiased agents relies on the event $\mathcal{E}$ that all initial samples of arm 1 (*i.e.,* the good arm) are realized as 0. This would indeed imply a 0-sampling failure (as long as at least one initial sample of arm 1 is realized to 1), but the event $\mathcal{E}$ happens with probability exponential in $N_0$, the number of initial samples. In contrast, in our result $p_{\texttt{fail}}$ only depends on $N_0$ through the assumption that $\Delta < O\left(1/\sqrt{N_0}\right)$.

*Discussion* 3.8. Corollary 3.6 can be seen as a general result on the failure of the greedy algorithm. This is the first such result with a non-trivial dependence on $N_0$, to the best of our knowledge.

We can remove assumption (3.2) and allow a small $N_0$ if the behavioral type for each agent $t$ also satisfies natural (and very mild) properties of symmetry and monotonicity:

- (P1) *(symmetry)* if all rewards in $\texttt{hist}_t$ are 0, the two arms are treated symmetrically;[10]

- (P2) *(monotonicity)* Fix any arm $a \in [2]$, any $t$-round history $H$ in which all rewards are 0 for both arms, and any other $t$-round history $H'$ that contains the same number of samples of arm $a$ such that all these samples have reward 1. Then

$$\Pr[a_t = a \mid \texttt{hist}_t = H'] \geq \Pr[a_t = a \mid \texttt{hist}_t = H]. \tag{3.6}$$

Note that both properties would still be natural and mild even without the "all rewards are zero" clause. The resulting guarantee on the failure probability is somewhat cleaner.

**Theorem 3.9** (small $N_0$). *Fix $\eta \geq 0$, assume Eq. (3.1), and let $N_0 \in [1, N^*]$, where $N^* := \lceil 64\eta/c^2 + 1/c \rceil$. Suppose each agent $t$ is $\eta$-confident and satisfies properties (P1) and (P2). Then an $n$-sampling failure, $n = N^* - N_0$, occurs with probability at least*

$$p_{\texttt{fail}} = \Omega_c(\ c^{2N^*}\ ) = \Omega_c(\ e^{-O_c(\eta)}\ ). \tag{3.7}$$

*Consequently,* $\text{Regret}(T) \geq \Delta \cdot p_{\texttt{fail}} \cdot (T - n)$.

If all agents are pessimistic, we find that *any levels of pessimism*, whether small or large or different across agents, lead to a 0-sampling failure with probability $\Omega_c(\Delta)$, matching Corollary 3.6 for the unbiased behavior. This happens in the (very reasonable) regime when

$$\Omega_c(\eta) < N_0 < O(1/\Delta^2). \tag{3.8}$$

**Theorem 3.10** (pessimistic agents). *Suppose each agent $t \in [T]$ is $\eta_t$-pessimistic, for some $\eta_t \geq 0$. Suppose assumptions (3.1) and (3.2) hold for $\eta = \max_{t \in [T]} \eta_t$. Then the 0-sampling failure occurs with probability lower-bounded by Eq. (3.5). Consequently,* $\text{Regret}(T) \geq \Omega_c(\Delta^2) \cdot e^{-O_c\left(N_0\,\Delta^2\right)}$.

We allow extremely pessimistic agents ($\eta_t \sim \log T$), and the pessimism levels $\eta_t$ can vary across agents $t$. While the relevant parameter is $\eta = \max_{t \in [T]} \eta_t$, the failure probability in (3.5) does not contain the $e^{-\eta}$ term. In particular, $p_{\texttt{fail}} = \Omega(\Delta)$ when $N_0 < O(1/\Delta^2)$. However, the dependence on $\eta$ "creeps in" through assumption (3.2), *i.e.,* that $N_0 > \Omega_c(\eta)$.

**Proof Overview.** We first show that the average reward of arm 1 (the good arm), is upper bounded by some threshold $q_1$. This is only guaranteed with some probability and only when this arm is sampled exactly $N$ times, for a particular $N \geq N_0$. Next, we lower bound the average reward of arm 2 (the bad arm): we show that with some probability it is *always* above some threshold $q_2 \in (q_1, \mu_2)$. Focus

---

[10]That is, the behavioral type stays the same if the arms' labels are switched.

on the round $t^*$ when the good arm is sampled for the $N$-th time (if this ever happens). If both of these events hold, from round $t^*$ onwards the bad arm will have a larger average reward by a constant margin $q_2 - q_1$. Consequently, as we prove, the bad arm has a larger index, and therefore gets chosen.

The details of this argument differ from one theorem to another. For Theorem 3.1, it suffices to set the thresholds $q_2, q_1$ such that $q_2 - q_1 = \Theta(\sqrt{\eta/N_0})$. For Theorem 3.10, we use a more involved argument: since the LCB of an arm increases when it is played, playing this arm only strengthens the preference of pessimistic agents for this arm. We are therefore less constrained in the choice of $q_1, q_2$ and we can prove that a learning failure occurs whenever $q_1 < q_2$.[11] In both proofs, we also require $q_1$ and $q_2$ to be close to $\mu_1$ and $\mu_2$, resp., so as to lower-bound the probability of the two desirable events. For Theorem 3.9, the case of small $N_0$, our analysis becomes more subtle. We can (in some sense) simplify the two events defined above, but we need to introduce a *third* event: if arm 1 is chosen by at least $n$ agents (for a suitably defined $n$), then arm 2 is chosen by $n$ agents before arm 1 is. The crux is a "deterministic" argument which derives a failure when all three events hold jointly.

To formalize, we represent realized rewards of each arm $a$ as written out in advance on a "tape", where each entry is an independent Bernoulli draw with mean $\mu_a$.[12] The $i$-th entry is returned as reward when and if arm $a$ is chosen for the $i$-th time. (We start counting from the initial samples, which comprise entries $i \in [N_0]$.) We analyze each arm separately (and then invoke independence).

We use some tools from probability: a sharp anti-concentration inequality for arm 1 and a martingale argument for arm 2. Let $(X_i)_{i \in \mathbb{N}}$ be a sequence of i.i.d. Bernoulli random variables with mean $p \in [c, 1-c]$, for some absolute constant $c \in (0, 1/2)$. The anti-concentration is as follows:

$$(\forall n \geq 1/c, \ q \in (c/8, p)) \qquad \Pr\left[\tfrac{1}{n} \sum_{i=1}^n X_i \leq q\right] \geq \Omega\left(e^{-O\left(n(p-q)^2\right)}\right), \qquad (3.9)$$

The martingale argument leads to this:

$$\forall q \in [0, p) \qquad \Pr\left[\forall n \geq 1: \quad \tfrac{1}{n} \sum_{i=1}^n X_i \geq q\right] \geq \Omega_c(p-q). \qquad (3.10)$$

We each tool to the tape for the respective arm, and lower bound the probability of the desirable event.

While the novelty is mainly in how we *use* these tools, the tools themselves are not very standard. Eq. (C.1) follows from the anti-concentration inequality in [61] and a reverse Pinsker inequality in [28]. More standard anti-concentration results via Stirling's approximation lead to an additional factor of $1/\sqrt{n}$ on the right-hand side of (C.1). For Eq. (C.2), we introduce an exponential martingale and relate the event in Eq. (C.2) to a deviation of this martingale. We then use Ville's inequality (a version of Doob's martingale inequality) to bound the probability that this deviation occurs.

# 4 Upper bounds for optimistic agents

We upper-bound regret for optimistic agents: we match the exponential-in-$\eta$ scaling from Corollary 3.4 and then extend this result to different behavioral types. On a technical level, we prove three regret bounds of the same shape (4.1), but with a different $\Phi$ term. (The unified presentation emphasizes this similarity.) Throughout, $\Delta = \mu_1 - \mu_2$ denotes the gap between the two arms.

**Theorem 4.1.** *Suppose all agents are $\eta$-optimistic, for some fixed $\eta > 0$. Then, letting $\Phi = \eta$,*

$$\text{Regret}(T) \leq O\left(T \cdot e^{-\Omega(\eta)} \cdot \Delta(1 + \log 1/\Delta) \ + \ \Phi/\Delta\right). \qquad (4.1)$$

*Discussion* 4.2. The main take-away is that the exponential-in-$\eta$ scaling from Corollary 3.4 is tight for $\eta$-optimistic agents, and therefore the best possible lower bound for $\eta$-confident agents. This result holds for any given $N_0$, the number of initial samples.[13] Our guarantee remains optimal in the "extreme optimism" regime when $\eta \sim \log(T)$, matching the optimal regret rate, $O(\log(T)/\Delta)$.

What if different agents can hold different behavioral types? First, let us allow agents to have varying amounts of optimism, possibly different across arms and possibly randomized.

**Definition 4.3.** Fix $\eta_{\max} \geq \eta > 0$. An agent $t \in [T]$ is called $[\eta, \eta_{\max}]$-*optimistic* if its index $\text{Ind}_{a,t}$ lies in the interval $\left[\text{UCB}_{a,t}^\eta, \text{UCB}_{a,t}^{\eta_{\max}}\right]$, for each arm $a \in [2]$.

---

[11]We also require $q_1 > \sqrt{\eta/N_0}$ to ensure that the confidence lower bounds are not truncated to zero.

[12]This is an equivalent (and well-known) representation of rewards in stochastic bandits.

[13]For ease of exposition, we do not track the improvements in regret when $N_0$ becomes larger.

We show that the guarantee in Theorem 4.1 is robust to varying the optimism level "upwards".

**Theorem 4.4** (robustness). *Fix $\eta_{\max} \geq \eta > 0$. Suppose all agents are $[\eta, \eta_{\max}]$-optimistic. Then regret bound (4.1) holds with $\Phi = \eta_{\max}$.*

Note that the upper bound $\eta_{\max}$ has only a mild influence on the regret bound in Theorem 4.4.

Our most general result only requires a small fraction of agents to be optimistic, whereas all agents are only required to be $\eta_{\max}$-confident (allowing all behaviors consistent with that).

**Theorem 4.5** (recurring optimism). *Fix $\eta_{\max} \geq \eta > 0$. Suppose all agents are $\eta_{\max}$-confident. Further, suppose each agent's behavioral type is chosen independently at random so that the agent is $[\eta, \eta_{\max}]$-optimistic with probability at least $q > 0$. Then regret bound (4.1) holds with $\Phi = \eta_{\max}/q$.*

Thus, with even a small fraction of optimists, $q > \frac{1}{\Delta \cdot o(T)}$, the behavioral type of less optimistic agents does not have much impact on regret. In particular, it does not hurt much if they become very pessimistic. **A small fraction of optimists goes a long way!** Further, a small-but-constant fraction of *extreme* optimists, *i.e.,* $\eta, \eta_{\max} \sim \log(T)$ in Theorem 4.5, yields optimal regret rate, $\log(T)/\Delta$.

## 5 Learning failures for Bayesian agents

In this section, we posit that agents are endowed with Bayesian beliefs. The basic version is that all agents believe that the mean reward of each arm is initially drawn from a uniform distribution on $[0, 1]$. (We emphasize that the mean rewards are fixed and *not* actually drawn according to these beliefs.) Each agent $t$ computes a posterior $\mathcal{P}_{a,t}$ for $\mu_a$ given the history $\texttt{hist}_t$, for each arm $a \in [a]$, and maps this posterior to the index $\texttt{Ind}_{a,t}$ for this arm.[14]

The basic behavior is that $\texttt{Ind}_{a,t}$ is the posterior mean reward, $\mathbb{E}[\mathcal{P}_{a,t}]$. We call such agents *Bayes-unbiased*. Further, we consider a Bayesian version of $\eta$-confident agents, defined by

$$\texttt{Ind}_{a,t} \in [Q_{a,t}(\zeta), Q_{a,t}(1 - \zeta)] \quad \text{for each arm } a \in [2], \tag{5.1}$$

where $Q_{a,t}(\cdot)$ denotes the quantile function of the posterior $\mathcal{P}_{a,t}$ and $\zeta \in (0, 1/2)$ is a fixed parameter (analogous to $\eta$ elsewhere). The interval in Eq. (5.1) is a Bayesian version of $\eta$-confidence intervals. Agents $t$ that satisfy Eq. (5.1) are called *$\zeta$-Bayes-confident*.

We allow more general beliefs given by independent Beta distributions. For each arm $a \in [2]$, all agents believe that the mean reward $\mu_a$ is initially drawn as an independent sample from Beta distribution with parameters $\alpha_a, \beta_a \in \mathbb{N}$. Our results are driven by parameter $M = \max_{a \in [2]} \alpha_a + \beta_a$. We refer to such beliefs as *Beta-beliefs with strength $M$*. The intuition is that the prior on each arm $a$ can be interpreted as being "based on" $\alpha_a + \beta_a - 2$ samples from this arm.[15]

Our technical contribution here is that Bayes-unbiased (resp., $\zeta$-Bayes-confident) agents are $\eta$-confident for a suitably large $\eta$, and therefore subject to the learning failure in Theorem 3.1.

**Theorem 5.1.** *Consider a Bayesian agent that holds Beta-beliefs with strength $M \geq 1$.*

> *(a) If the agent is Bayes-unbiased, then it is $\eta$-confident for some $\eta = O(M/\sqrt{N_0})$.*
> *(b) If the agent is $\zeta$-Bayes-confident, then it is $\eta$-confident for $\eta = O\left(M/\sqrt{N_0} + \ln(1/\zeta)\right)$.*

*Discussion* 5.2. Beta-beliefs may be completely unrelated to the actual mean rewards. If $\zeta$ and $M$ are constants relative to $T$, the resulting $\eta$ is constant, too. Our guarantee is stronger if the beliefs are weak (*i.e.,* $M$ is small) or are "dominated" by the initial samples, in the sense that $N_0 > \Omega(M^2)$.

*Discussion* 5.3. $\zeta$-Bayes-confident agents subsume Bayesian version of optimism and pessimism, where the index $\texttt{Ind}_{a,t}$ is defined as, resp., $Q_{a,t}(1 - \zeta)$ and $Q_{a,t}(\zeta)$, as well as all the Bayesian versions of all other behaviorial biases discussed previously as special cases of $\eta$-confidence.

## 6 Bayesian model with arbitrary priors

We consider Bayesian-unbiased agents in a "fully Bayesian" model such that the mean rewards are actually drawn from a prior. We are interested in *Bayesian probability* and *Bayesian regret*, *i.e.,* resp.,

---

[14]Note that the Bayesian update for agent $t$ does not depend on the beliefs of the previous agents.

[15]More precisely, any Beta distribution with integer parameters $(\alpha, \beta)$ can be seen as a Bayesian posterior obtained by updating a uniform prior on $[0, 1]$ with $\alpha + \beta - 2$ data points.

probability and regret in expectation over the prior. We focus on learning failures when the agents never choose an arm with the largest prior mean reward (as opposed to an arm with the largest *realized* mean reward, which is not necessarily the same arm). We do not explicitly allow initial samples (*i.e.,* we posit $N_0 = 0$ here), because they are implicitly included in the prior.

Compared to Section 5, the benefit is that we allow arbitrary priors, possibly correlated across the two arms. Further, our guarantee does not depend on the prior, other than through the *prior gap* $\mathbb{E}[\mu_1 - \mu_2]$, and does not contain any hidden constants. On the other hand, the guarantees here are only in expectation over the prior, whereas the ones in Section 5 hold for fixed $\mu_1, \mu_2$. Also, our result here is restricted to Bayesian-unbiased agents.

**Theorem 6.1.** *Suppose the pair $(\mu_1, \mu_2)$ is initially drawn from some Bayesian prior $\mathcal{P}$ such that $\mathbb{E}[\mu_1] > \mathbb{E}[\mu_2]$. Assume that all agents are Bayesian-unbiased, with beliefs given by $\mathcal{P}$. Then with Bayesian probability at least $\mathbb{E}[\mu_1 - \mu_2]$, the agents never choose arm $2$.*

*Proof.* W.l.o.g., assume that agents break ties in favor of arm 2. In each round $t$, the key quantity is $Z_t = \mathbb{E}[\mu_1 - \mu_2 \mid \mathtt{hist}_t]$. Indeed, arm 2 is chosen if and only if $Z_t \leq 0$. Let $\tau$ be the first round when arm 2 is chosen, or $T + 1$ if this never happens. We use martingale techniques to prove that

$$\mathbb{E}[Z_\tau] = \mathbb{E}[\mu_1 - \mu_2]. \tag{6.1}$$

We use the optional stopping theorem (OST). We observe that $\tau$ is a stopping time relative to $\mathcal{H} = (\, \mathtt{hist}_t : t \in [T + 1] \,)$, and $(\, Z_t : t \in [T + 1] \,)$ is a martingale relative to $\mathcal{H}$. [16] OST asserts that $\mathbb{E}[Z_\tau] = \mathbb{E}[Z_1]$ for any martingale $Z_t$ and any bounded stopping time $\tau$. Eq. (6.1) follows because $\mathbb{E}[Z_1] = \mathbb{E}[\mu_1 - \mu_2]$. On the other hand, by Bayes' theorem it holds that

$$\mathbb{E}[Z_\tau] = \Pr\left[\tau \leq T\right] \mathbb{E}[Z_\tau \mid \tau \leq T] + \Pr\left[\tau > T\right] \mathbb{E}[Z_\tau \mid \tau > T] \tag{6.2}$$

Recall that $\tau \leq T$ implies that arm 2 is chosen in round $\tau$, which in turn implies that $Z_\tau \leq 0$. It follows that $\mathbb{E}[Z_\tau \mid \tau \leq T] \leq 0$. Plugging this into Eq. (6.2), we find that

$$\mathbb{E}[\mu_1 - \mu_2] = \mathbb{E}[Z_\tau] \leq \Pr\left[\tau > T\right] = \Pr\left[\text{arm 2 never chosen}\right]. \qquad \square$$

As a corollary, we derive a 0-sampling failure, leading to $\Omega(T)$ Bayesian regret. Specifically, the agents start out playing arm 1 (because it has a higher prior mean reward), and never try arm 2 *when it is in fact the best arm*. This happens whenever the prior is independent across arms and has a positive density on the entire $[0, 1]$ interval (see the supplement for the exact statement). Note that it is a (much) more general family of priors compared to independent Beta-priors allowed in Section 5.

# 7 Conclusions and open questions

We examine the dynamics of social learning in a multi-armed bandit scenario, where agents sequentially choose arms and receive rewards, and observe the full history of previous agents. For a range of agents' myopic behavior, we investigate how they impact exploration, and provide tight upper and lower bounds on the learning failure probabilities and regret rates. In particular, we obtain the first general results on the failure of the greedy algorithm in bandits.

With our results as a "departure point", one could study BSL in more complex bandit models with many arms and/or some known structure of rewards that the agents' myopic behaviour would account for.[17] The greedy algorithm fails for some structures (*e.g.,* our current model) and works well for some others (*e.g.,* for linear contextual bandits with smoothed contexts [37, 11, 51], or when all arms have the same rewards). The whole world is in between these two extremes. It is not at all clear which structures would cause learning failures and which would enable learning, and which structures would be amenable to analysis, one way or another.

# 8 Acknowledgements

This work is partially supported by DARPA QuICC NSF and AF:Small #2218678, #2114269.

---

[16]The latter follows from a general fact that sequence $\mathbb{E}[X \mid \mathtt{hist}_t], t \in [T + 1]$ is a martingale w.r.t. $\mathcal{H}$ for any random variable $X$ with $\mathbb{E}[\,|X|\,] < \infty$. It is known as *Doob martingale* for $X$.

[17]*E.g.,* linear, convex, Lipschitz and combinatorial structures are well-studied, see books [17, 54, 45].

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

## SUPPLEMENTARY MATERIALS FOR "BAYESIAN BANDIT SOCIAL LEARNING UNDER MYOPIC BEHAVIOR"

The supplement contains the proofs, as well as detailed discussion of related work on social learning.

## Contents

# A Related Work on Social Learning

A vast literature on social learning studies agents that learn over time in a shared environment. A prominent topic is the presence or absence of learning failures such as ours. Models vary across several dimensions, such as: which information is acquired or transmitted, what is the communication network, whether agents are long-lived or only act once, how they choose their actions, etc. Below we discuss several lines of work that are most relevant.

In "sequential social learning", starting from [9, 60, 14, 56], agents observe private signals, but only the chosen actions are observable in the future; see Golub and Sadler [27] for a survey. The social planner (who chooses agents' actions given access to the knowledge of all previous agents) only needs to *exploit*, *i.e.,* choose the best action given the previous agents' signals, whereas in our model it also needs to *explore*. Learning failures are (also) of primary interest, but they occur for an entirely different reason: restricted information flow, *i.e.,* the fact that the private signals are not observable in the future.

"Strategic experimentation", starting from Bolton and Harris [16] and Keller et al. [39], studies long-lived learning agents that observe both actions and rewards of one another; see Hörner and Skrzypacz [33] for a survey. Here, the social planner also solves a version of multi-armed bandits, albeit a very different one (with time-discounting, "safe" arm that is completely known, and "risky" arm that follows a stochastic process). The main difference is that the agents engage in a complex repeated game where they explore but prefer to free-ride on exploration by others.

Bala and Goyal [8] and Lazer and Friedman [46] consider a network of myopic learners, all faced with the same bandit problem and observing each other's actions and rewards. The interaction protocol is very different from ours: agents are long-lived, act all at once, and only observe their neighbors on the network. Other specifics are different, too. Bala and Goyal [8] makes strong assumptions on learners' beliefs, which would essentially cause the greedy algorithm to work well in BSL. In Lazer and Friedman [46], each learner only retains the best observed action, rather than the full history. Both papers study social learning under different network topologies.

Prominent recent work, *e.g.,* [31, 15, 25, 44], targets agents with *misspecified beliefs*, *i.e.,* beliefs whose support does not include the correct model. The framing is similar to BSL with Bayesian-unbiased agents: agents arrive one by one and face the same decision problem, whereby each agent makes a rational decision after observing the outcomes of the previous agents.[18] Rational decisions under misspecified beliefs make a big difference compared to BSL, and structural assumptions about rewards/observations and the state space tend to be very different from ours. The technical questions being asked tend to be different, too. *E.g.,* convergence of beliefs is of primary interest, whereas the chosen arms and agents' beliefs/estimates trivially converge in our setting. [19]

# B Preliminaries: Reward-Tape

It is convenient for our analyses to interpret the realized rewards of each arm as if they are written out in advance on a "tape". We posit a matrix $\left( \texttt{Tape}_{a,i} \in [0,1] : a \in [2], i \in [T] \right)$, called *reward-tape*, such that each entry $\texttt{Tape}_{a,i}$ is an independent Bernoulli draw with mean $\mu_a$. This entry is returned as reward when and if arm $a$ is chosen for the $i$-th time. (We start counting from the initial samples, which comprise entries $i \in [N_0]$.) This is an equivalent (and well-known) representation of rewards in stochastic bandits.

We will use the notation for the UCBs/LCBs defined by the reward-tape. Fix arm $a \in [2]$ and $n \in [T]$. Let $\widehat{\mu}_{a,n}^{\texttt{tape}} = \frac{1}{n} \sum_{i \in [n]} \texttt{Tape}_{a,i}$ be the average over the first $n$ entries for arm $a$. Now, given $\eta \geq 0$, define the appropriate confidence bounds:

$$\texttt{UCB}_{a,n}^{\texttt{tape},\,\eta} := \min\left\{ 1, \widehat{\mu}_{a,n}^{\texttt{tape}} + \sqrt{\eta/n} \right\} \quad \text{and} \quad \texttt{LCB}_{a,n}^{\texttt{tape},\,\eta} := \max\left\{ 0, \widehat{\mu}_{a,n}^{\texttt{tape}} - \sqrt{\eta/n} \right\}. \quad \text{(B.1)}$$

---

[18]The original framing in this work posits a single learner that makes (possibly) myopic decisions over time and observes their outcomes. An alternative interpretation is that each decision is made by a new myopic agent who observes the history.

[19]Essentially, if an arm is chosen infinitely often then the agents beliefs/estimates converge on its true mean reward; else, the agents eventually stop receiving any new information about this arm.

# C Proofs from Section 3: Learning Failures

Our proofs rely on two tools from Probability (proved in Section C.4 and C.5): a sharp anti-concentration inequality for Binomial distribution and a lemma that encapsulates a martingale argument.

**Lemma C.1** (anti-concentration). *Let $(X_i)_{i \in \mathbb{N}}$ be a sequence of independent Bernoulli random variables with mean $p \in [c, 1-c]$, for some $c \in (0, 1/2)$ interpreted as an absolute constant. Then*

$$(\forall n \geq 1/c, \ q \in (c/8, \, p)) \qquad \Pr\left[ \tfrac{1}{n} \sum_{i=1}^{n} X_i \leq q \right] \geq \Omega\left( e^{-O\left( n(p-q)^2 \right)} \right), \qquad \text{(C.1)}$$

*where $\Omega(\cdot)$ and $O(\cdot)$ hide the dependence on c.*

**Lemma C.2** (martingale argument). *In the setting of Lemma C.1,*

$$\forall q \in [0, p) \qquad \Pr\left[ \forall n \geq 1: \quad \tfrac{1}{n} \sum_{i=1}^{n} X_i \geq q \right] \geq \Omega_c(p - q). \qquad \text{(C.2)}$$

The overall argument will be as follows. We will use Lemma C.1 to upper-bound the average reward of arm 1, *i.e.,* the good arm, by some threshold $q_1$. This upper bound will only be guaranteed to hold when this arm is sampled exactly $N$ times, for a particular $N \geq N_0$. Lemma C.2 will allow us to uniformly *lower*-bound the average reward of arm 2, *i.e.,* the bad arm, by some threshold $q_2 \in (q_1, \mu_2)$. Focus on the round $t^*$ when the good arm is sampled for the $N$-th time (if this ever happens). If the events in both lemmas hold, from round $t^*$ onwards the bad arm will have a larger average reward by a constant margin $q_2 - q_1$. We will prove that this implies that the bad arm has a larger index, and therefore gets chosen by the agents. The details of this argument differ from one theorem to another.

Lemma C.1 is a somewhat non-standard statement which follows from the anti-concentration inequality in [61] and a reverse Pinsker inequality in [28]. More standard anti-concentration results via Stirling's approximation lead to an additional factor of $1/\sqrt{n}$ on the right-hand side of (C.1). For Lemma C.2, we introduce an exponential martingale and relate the event in (C.2) to a deviation of this martingale. We then use Ville's inequality (a version of Doob's martingale inequality) to bound the probability that this deviation occurs.

## C.1 Proof of Theorem 3.1: $\eta$-confident agents

Fix thresholds $q_1 < q_2$ to be specified later. Define two "failure events":

$\mathtt{Fail}_1$: the average reward of arm 1 after the $N_0$ initial samples is below $q_1$;

$\mathtt{Fail}_2$: the average reward of arm 2 is never below $q_2$.

In a formula, using the reward-tape notation from Appendix B, these events are

$$\mathtt{Fail}_1 := \left\{ \widehat{\mu}_{1, N_0}^{\mathtt{tape}} \leq q_1 \right\} \quad \text{and} \quad \mathtt{Fail}_2 := \left\{ \forall n \in [T] : \widehat{\mu}_{2, n}^{\mathtt{tape}} \geq q_2 \right\}. \qquad \text{(C.3)}$$

We show that event $\mathtt{Fail} := \mathtt{Fail}_1 \cap \mathtt{Fail}_2$ implies the 0-sampling failure, as long as the margin $q_2 - q_1$ is sufficiently large.

**Claim C.3.** *Assume $q_2 - q_1 > 2 \cdot \sqrt{\eta/N_0}$ and event $\mathtt{Fail}$. Then arm 1 is never chosen by the agents.*

*Proof.* Assume, for the sake of contradiction, that some agent chooses arm 1. Let $t$ be the first round when this happens. Note that $\mathtt{Ind}_{1,t} \geq \mathtt{Ind}_{2,t}$. We will show that this is not possible by upper-bounding $\mathtt{Ind}_{1,t}$ and lower-bounding $\mathtt{Ind}_{2,t}$.

By definition of round $t$, arm 1 has been previously sampled exactly $N_0$ times. Therefore,

$$\begin{aligned}
\mathtt{Ind}_{1,t} &\leq \widehat{\mu}_{1, N_0}^{\mathtt{tape}} + \sqrt{\eta/N_0} && \textit{(by definition of index)} \\
&\leq q_1 + \sqrt{\eta/N_0} && \textit{(by $\mathtt{Fail}_1$)} \\
&< q_2 - \sqrt{\eta/N_0} && \textit{(by assumption)}.
\end{aligned}$$

Let $n$ be the number of times arm 2 has been sampled before round $t$. This includes the initial samples, so $n \geq N_0$. It follows that

$$\mathtt{Ind}_{2,t} \geq \widehat{\mu}_{2,n}^{\mathtt{tape}} - \sqrt{\eta/n} \qquad \textit{(by definition of index)}$$

$$\geq q_2 - \sqrt{\eta/N_0} \qquad \textit{(by } \mathtt{Fail}_2 \textit{ and } n \geq N_0 \textit{)}.$$

Consequently, $\mathtt{Ind}_{2,t} > \mathtt{Ind}_{1,t}$, contradiction. $\qquad \square$

In what follows, let $c$ be the absolute constant from assumption (3.1).

Let us lower bound $\Pr[\mathtt{Fail}]$ by applying Lemmas C.1 and C.2 to the reward-tape.

**Claim C.4.** *Assume $c/4 < q_1 < q_2 < \mu_2$. Then*

$$\Pr[\mathtt{Fail}] \geq q_{\mathtt{fail}} := \Omega_c(\mu_2 - q_2) \cdot e^{-O_c\left(N_0(\mu_1 - q_1)^2\right)}. \qquad (C.4)$$

*Proof.* To handle $\mathtt{Fail}_1$, apply Lemma C.1 to the reward-tape for arm 1, *i.e.*, to the random sequence $(\mathtt{Tape}_{1,i})_{i \in [T]}$, with $n = N_0$ and $q = q_1$. Recalling that $N_0 \geq 1/c$ by assumption (3.2),

$$\Pr[\mathtt{Fail}_1] \geq \Omega_c\left(e^{-O_c\left(N_0(\mu_1 - q_1)^2\right)}\right).$$

To handle $\mathtt{Fail}_2$, apply Lemma C.2 to the reward-tape for arm 2, *i.e.*, to the random sequence $(\mathtt{Tape}_{2,i})_{i \in [T]}$, with threshold $q = q_2$. Then

$$\Pr[\mathtt{Fail}_2] \geq \Omega_c(\mu_2 - q_2).$$

Events $\mathtt{Fail}_1$ and $\mathtt{Fail}_2$ are independent, because they are determined by, resp., realized rewards of arm 1 and realized rewards of arm 2. The claim follows. $\qquad \square$

Finally, let us specify suitable thresholds that satisfy the preconditions in Claims C.3 and C.4:

$$q_1 := \mu_2 - 4 \cdot \sqrt{\eta/N_0} - c\Delta/4 \quad \text{and} \quad q_2 := \mu_2 - \sqrt{\eta/N_0} - c\Delta/4.$$

Plugging in $\mu_2 \geq c$ and $N_0 \geq 64 \cdot \eta/c^2$, it is easy to check that $q_1 \geq c/4$, as needed for Claim C.4.

Thus, the preconditions in Claims C.3 and C.4 are satisfied. It follows that the 0-failure happens with probability at least $q_{\mathtt{fail}}$, as defined in Claim C.4. We obtain the final expression in Eq. (3.3) because $\mu_a - q_a = \Theta_c(\Delta + \sqrt{\eta/N_0})$ for both arms $a \in [2]$.

## C.2 Proof of Theorem 3.10: pessimistic agents

We reuse the machinery from Section C.1: we define event $\mathtt{Fail} := \mathtt{Fail}_1 \cap \mathtt{Fail}_2$ as per Eq. (C.3), for some thresholds $q_1 < q_2$ to be specified later, and use Claim C.4 to bound $\Pr[\mathtt{Fail}]$. However, we need a different argument to prove that $\mathtt{Fail}$ implies the 0-sampling failure, and a different way to set the thresholds.

**Claim C.5.** *Assume $q_1 > \sqrt{\eta/N_0}$ and event $\mathtt{Fail}$. Then arm 1 is never chosen by the agents.*

*Proof.* Assume, for the sake of contradiction, that some agent chooses arm 1. Let $t$ be the first round when this happens. Note that $\mathtt{Ind}_{1,t} \geq \mathtt{Ind}_{2,t}$. We will show that this is not possible by upper-bounding $\mathtt{Ind}_{1,t}$ and lower-bounding $\mathtt{Ind}_{2,t}$.

By definition of round $t$, arm 1 has been previously sampled exactly $N_0$ times. Therefore,

$$\mathtt{Ind}_{1,t} = \max\{0, \widehat{\mu}_{1,N_0}^{\mathtt{tape}} - \sqrt{\eta/N_0}\} \qquad \textit{(by definition of index)}$$

$$\leq \max\{0, q_1 - \sqrt{\eta/N_0}\} \qquad \textit{(by } \mathtt{Fail}_1 \textit{)}$$

$$= q_1 - \sqrt{\eta/N_0} \qquad \textit{(by assumption)}.$$

Let $n$ be the number of times arm 2 has been sampled before round $t$. This includes the initial samples, so $n \geq N_0$. It follows that

$$\mathtt{Ind}_{2,t} \geq \widehat{\mu}_{2,n}^{\mathtt{tape}} - \sqrt{\eta/n} \qquad \textit{(by definition of index)}$$

$$\geq q_2 - \sqrt{\eta/N_0} \qquad \textit{(by } \mathtt{Fail}_2 \textit{ and } n \geq N_0 \textit{)}.$$

Consequently, $\mathtt{Ind}_{2,t} > \mathtt{Ind}_{1,t}$, contradiction. $\qquad \square$

Now, set the thresholds $q_1, q_2$ as follows:

$$q_1 := \mu_2 - c\Delta/4 \quad \text{and} \quad q_2 := \mu_2 - c\Delta/8.$$

Plugging in $\mu_2 \geq c$ and $N_0 \geq 64 \cdot \eta/c^2$, it is easy to check that the preconditions in Claims C.4 and C.5 are satisfied. So, the 0-failure happens with probability at least $q_{\texttt{fail}}$ from Claim C.4. The final expression in Eq. (3.3) follows because $\mu_a - q_a = \Theta_c(\Delta)$ for both arms $a \in [2]$.

### C.3 Proof of Theorem 3.9: small $N_0$

We focus on the case when $N_0 \leq N^* := \lceil 64\eta/c^2 + 1/c \rceil$. We can now afford to handle the initial samples in a very crude way: our failure events posit that all initial samples of the good arm return reward 0, and all initial samples of the bad arm return reward 1.

$$\texttt{Fail}_1 := \left\{ \forall i \in [1, N^*] : \texttt{Tape}_{1,i} = 0 \right\},$$
$$\texttt{Fail}_2 := \left\{ \forall i \in [1, N^*] : \texttt{Tape}_{2,i} = 1 \quad \text{and} \quad \forall i \in [T] : \widehat{\mu}_{2,i}^{\texttt{tape}} \geq q_2 \right\}.$$

Here, $q_2 > 0$ is the threshold to be defined later.

On the other hand, our analysis given these events becomes more subtle. In particular, we introduce another "failure event" $\texttt{Fail}_3$, with a more subtle definition: if arm 1 is chosen by at least $n := N^* - N_0$ agents, then arm 2 is chosen by $n$ agents before arm 1 is.

We first show that $\texttt{Fail} := \texttt{Fail}_1 \cap \texttt{Fail}_2 \cap \texttt{Fail}_3$ implies the $n$-sampling failure.

**Claim C.6.** *Assume that $q_2 \geq c/4$ and $\texttt{Fail}$ holds. Then at most $n = N^* - N_0$ agents choose arm 1.*

*Proof.* For the sake of contradiction, suppose arm 1 is chosen by more than $n$ agents. Let agent $t$ be the $(n+1)$-th agent that chooses arm 1. In particular, $\texttt{Ind}_{1,t} \geq \texttt{Ind}_{2,t}$.

By definition of $t$, arm 1 has been previously sampled exactly $N^*$ times before (counting the $N_0$ initial samples). Therefore,

$$
\begin{aligned}
\texttt{Ind}_{1,t} &\leq \widehat{\mu}_{1,N^*}^{\texttt{tape}} + \sqrt{\eta/N^*} && \textit{(by $\eta$-confidence)} \\
&= \sqrt{\eta/N^*} && \textit{(by event $\texttt{Fail}_1$)} \\
&\leq c/8 && \textit{(by definition of $N^*$).}
\end{aligned}
$$

Let $m$ be the number of times arm 2 has been sampled before round $t$. Then

$$
\begin{aligned}
\texttt{Ind}_{2,t} &\geq \widehat{\mu}_{2,m}^{\texttt{tape}} - \sqrt{\eta/m} && \textit{(by $\eta$-confidence)} \\
&\geq q_2 - \sqrt{\eta/m} && \textit{(by event $\texttt{Fail}_2$)} \\
&\geq q_2 - \sqrt{\eta/N^*} && \textit{(since $m \geq N^*$ by event $\texttt{Fail}_3$)} \\
&\geq q_2 - c/8 && \textit{(by definition of $N^*$)} \\
&> c/8 && \textit{(since $q_2 \geq c/2$).}
\end{aligned}
$$

Therefore, $\texttt{Ind}_{2,t} > \texttt{Ind}_{1,t}$, contradiction. $\qquad\square$

Next, we lower bound the probability of $\texttt{Fail}_1 \cap \texttt{Fail}_2$ using Lemma C.2.

**Claim C.7.** *If $q_2 < \mu_2$ then $\Pr[\texttt{Fail}_1 \cap \texttt{Fail}_2] \geq \Omega_c(\mu_2 - q_2) \cdot c^{2\,N^*}$.*

*Proof.* Instead of analyzing $\texttt{Fail}_2$ directly, consider events

$$\mathcal{E} := \left\{ \forall i \in [1, N^*] : \texttt{Tape}_{2,i} = 1 \right\} \text{ and } \mathcal{E}' := \left\{ \forall m \in [N^* + 1, T] : \frac{1}{m-N^*} \sum_{i=N^*+1}^{m} \texttt{Tape}_{2,i} \geq q_2 \right\}.$$

Note that $\mathcal{E} \cap \mathcal{E}'$ implies $\texttt{Fail}_2$. Now, $\Pr[\texttt{Fail}_1] \geq \mu_1^{N^*} \geq c^{N^*}$ and $\Pr[\mathcal{E}] \geq (1-\mu_2)^{N^*} \geq c^{N^*}$. Further, $\Pr[\mathcal{E}'] \geq \Omega_c(\mu_2 - q_2)$ by Lemma C.2. The claim follows since these three events are mutually independent. $\qquad\square$

To bound $\Pr[\texttt{Fail}]$, we argue indirectly, assuming $\texttt{Fail}_1 \cap \texttt{Fail}_2$ and proving that the conditional probability of $\texttt{Fail}_3$ is at least $^1/_2$. While this statement feels natural given that $\texttt{Fail}_1 \cap \texttt{Fail}_2$ favors arm 2, the proof requires a somewhat subtle inductive argument. This is where we use the symmetry and monotonicity properties from the theorem statement.

**Claim C.8.** $\Pr[\texttt{Fail}_3 \mid \texttt{Fail}_1 \cap \texttt{Fail}_2] \geq \frac{1}{2}$.

Now, we can lower-bound $\Pr[\texttt{Fail}]$ by $\Omega_c(\mu_2 - q_2) \cdot c^{2N^*}$. Finally, we set the threshold to $q_2 = c/2$ and the theorem follows.

*Proof of Claim C.8.* Note that event $\texttt{Fail}_t$ is determined by the first $N^*$ entries of the reward-tape for both arms, in the sense that it does not depend on the rest of the reward-tape.

For each arm $a$ and $i \in [T]$, let agent $\tau_{a,i}$ be the $i$-th agent that chooses arm $a$, if such agent exists, and $\tau_i = T + 1$ otherwise. Then

$$\texttt{Fail}_3 = \{\, \tau_{2,n} \leq \tau_{1,n} \,\} = \{\, \tau_{1,n} \geq 2n \,\} \tag{C.5}$$

Let $\mathcal{E}$ be the event that the first $N^*$ entries of the reward-tape are $0$ for both arms. By symmetry between the two arms (property (P1) in the theorem statement) we have

$$\Pr[\tau_{2,n} < \tau_{1,n} \mid \mathcal{E}] = \Pr[\tau_{2,n} > \tau_{1,n} \mid \mathcal{E}] = {}^1/_2,$$

and therefore

$$\Pr[\texttt{Fail}_3 \mid \mathcal{E}] = \Pr[\tau_{2,n} \leq \tau_{1,n} \mid \mathcal{E}] \geq {}^1/_2. \tag{C.6}$$

Next, for two distributions $F, G$, write $F \succeq_{\texttt{fosd}} G$ if $F$ first-order stochastically dominates $G$. A conditional distribution of random variable $X$ given event $\mathcal{E}$ is denoted $(X|\mathcal{E})$. For each $i \in [T]$, we consider two conditional distributions for $\tau_{1,i}$: one given $\texttt{Fail}_1 \cap \texttt{Fail}_2$ and another given $\mathcal{E}$, and prove that the former dominates:

$$(\tau_{1,i} \mid \texttt{Fail}_1 \cap \texttt{Fail}_2) \succeq_{\texttt{fosd}} (\tau_{1,i} \mid \mathcal{E}) \quad \forall i \in [T]. \tag{C.7}$$

Applying (C.7) with $i = n$, it follows that

$$\begin{aligned}
\Pr[\texttt{Fail}_3 \mid \texttt{Fail}_1 \cap \texttt{Fail}_2] &= \Pr[\tau_{1,n} \geq 2n \mid \texttt{Fail}_1 \cap \texttt{Fail}_2] \\
&\geq \Pr[\tau_{1,n} \geq 2n \mid \mathcal{E}] = {}^1/_2.
\end{aligned}$$

(The last equality follows from (C.6) and Eq. (C.6).) Thus, it remains to prove (C.7).

Let us consider a fixed realization of each agents' behavioral type, *i.e.,* a fixed, deterministic mapping from histories to arms. W.l.o.g. interpret the behavioral type of each agent $t$ as first deterministically mapping history $\texttt{hist}_t$ to a number $p_t \in [0,1]$, then drawing a threshold $\theta_t \in [0,1]$ independently and uniformly at random, and then choosing arm 1 if and only if $p_t \geq \theta_t$. Note that $p_t = \Pr[a_t = 1 \mid \texttt{hist}_t]$. So, we pre-select the thresholds $\theta_t$ for each agent $t$. Note the agents retain the monotonicity property (P2) from the theorem statement. (For this property, the probabilities on both sides of Eq. (3.6) are now either $0$ or $1$.)

Let us prove (C.7) for this fixed realization of the types, using induction on $i$. Both sides of (C.7) are now deterministic; let $A_i, B_i$ denote, resp., the left-hand side and the right-hand side. So, we need to prove that $A_i \geq B_i$ for all $i \in [n]$. For the base case, take $i = 0$ and define $A_0 = B_0 = 0$. For the inductive step, assume $A_i \geq B_i$ for some $i \geq 0$. We'd like to prove that $A_{i+1} \geq B_{i+1}$. Suppose, for the sake of contradiction, that this is not the case, *i.e.,* $A_{i+1} < B_{i+1}$. Since $A_i < A_{i+1}$ by definition of the sequence $(\tau_{a,i} : \in [T])$, we must have

$$B_i \leq A_i < A_{i+1} < B_{i+1}.$$

Focus on round $t = A_{i+1}$. Note that the history $\texttt{hist}_t$ contains exactly $i$ agents that chose arm 1, both under event $\texttt{Fail}_1 \cap \texttt{Fail}_2$ and under event $\mathcal{E}$. Yet, arm 2 is chosen under $\mathcal{E}$, while arm 1 is chosen under $\texttt{Fail}_1 \cap \texttt{Fail}_2$. This violates the monotonicity property (P2) from the theorem statement. Thus, we've proved (C.7) for any fixed realization of the types. Consequently, (C.7) holds in general. $\square$

## C.4 Proof of Lemma C.1

*Proof.* We use the following sharp lower bound on the tail probability of binomial distribution.

**Theorem C.9** (Theorem 9 in [61]). *Let $n \in \mathbb{N}$ be a positive integer and let $(X_i)_{i \in [n]}$ be a sequence of i.i.d Bernoulli random variables with prameter p. For any $\beta > 1$ there exists constants $c_\beta$ and $C_\beta$ that only rely on $\beta$, such that for all $x$ satisfying $x \in [0, \frac{np}{\beta}]$ and $x + n(1-p) \geq 1$, we have*

$$\Pr\left[\sum_{i=1}^{n} X_i \leq np - x\right] \geq c_\beta e^{-C_\beta n D(p - \frac{x}{n} || p)},$$

*where $D(x||y)$ denotes the KL divergence between two Bernoulli random variables with parameters $x$ and $y$.*

We use the above result with $x = n(p - q)$ and $\beta = \frac{1-c}{1-\frac{9}{8}c}$. Note that $\beta > 1$ since $c < \frac{1}{2}$. We first verify that $x, \beta$ satisfy the conditions of the lemma. The $x + n(1-p) \geq 1$ condition holds by the assumption $n \geq 1/c$:

$$x + n(1-p) \geq n(1-p) \geq nc \geq 1.$$

As for the $x \leq \frac{np}{\beta}$ condition, by definition of $x$,

$$\frac{np}{x} = \frac{np}{n(p-q)} = \frac{p}{p-q}.$$

Since $p \leq 1 - c$ and $\frac{p}{p-q}$ is decreasing in $p$ for $p \geq q$, we can further bound this with

$$\frac{p}{p-q} \geq \frac{1-c}{1-c-q} \geq \frac{1-c}{1-c-\frac{c}{8}} = \beta,$$

where the second inequality follows from $q \geq c/8$ and $q < p \leq 1-c$, together with the fact that $\frac{1-c}{1-c-q}$ is decreasing in $q$ for $q < 1 - c$. We obtain $x \leq \frac{np}{\beta}$ by rearranging.

Invoking Theorem C.9 with the given values, we obtain

$$\Pr\left[\frac{\sum_{i=1}^{n} X_i}{n} \leq q\right] \geq c_\beta e^{-C_\beta n D(q||p)} = \Omega(e^{-O(nD(q||p))}). \tag{C.8}$$

Next, we use the following type of reverse Pinsker's inequality to upper bound $D(q||p)$.

**Theorem C.10** ([28]). *For any two probability measures $P$ and $Q$ on a finite support $X$, if $Q$ is absolutely continuous with respect to $P$, then the their KL divergence $D(Q||P)$ is upper bounded by $\frac{2}{\alpha_P} \delta(Q, P)^2$ where $\alpha_P = \min_{x \in X} P(x)$ and $\delta(Q, P)$ denotes the total variation distance between $P$ and $Q$.*

Setting $P = \text{Bernoulli}(p)$ and $Q = \text{Bernoulli}(q)$, we have $\alpha_P = \min(p, 1-p)$, and $\delta(Q, P) = p - q$ Therefore, since $\min(p, 1-p) \geq c$ by assumption, we conclude $D(q||p) \leq O((p-q)^2)$. Plugging this back in Equation (C.8) finshes the proof. $\square$

## C.5 Proof of Lemma C.2

Our proof will rely on the following doob-style inequality for (super)martingales.

**Lemma C.11** (Ville's Inequality [58]). *Let $(Z_n)_{n \geq 0}$ be a positive supermartingale with respect to filtration $(\mathcal{F}_n)_{n \geq 0}$, i.e. $Z_n \geq \mathbb{E}[Z_{n+1}|\mathcal{F}_n]$ for any $n \geq 0$. Then the following holds for any $x > 0$,*

$$\Pr\left[\max_{n \geq 0} Z_n \geq x\right] \leq \mathbb{E}[Z_0]/x.$$

In order to use this result, we will define the martingale $Z_n := u^{\sum_{i=1}^{n}(X_{i+1} - q)}$ for a suitable choice of $u$ as specified by the following lemma.

**Lemma C.12.** *Let c be an absolute constant. For any $p \in (c, 1 - c)$ and $q \in (0, p)$, there exists a value of $u \in (0, 1)$ such that*

$$(p \cdot u^{1-q} + (1 - p) \cdot u^{-q}) = 1. \tag{C.9}$$

*In addition, u satisfies*

$$p(1 - u^{1-q}) \geq \Omega(p - q). \tag{C.10}$$

*Proof.* To see why such a $u$ exists, define $f(x) = (p \cdot x^{1-q} + (1 - p) \cdot x^{-q})$. It is clear that $f(1) = 1$ and $\lim_{x \to 0} f(x) = \infty$ as $\lim_{x \to 0}(1 - p)x^{-q} = \infty$. Furthermore,

$$f'(x) = p \cdot (1 - q) \cdot x^{-q} + (1 - p) \cdot (-q) \cdot x^{-q-1},$$

which implies

$$f'(1) = p(1 - q) - (1 - p)q = p - q > 0.$$

Therefore, $f(x)$ is decreasing at $x = 1$. Since $\lim_{x \to 0} f(x) > f(1)$, this implies that $f(u) = f(1)$ for some $u \in (0, 1)$, proving Equation (C.9).

We now prove Equation (C.10), define $x_0$ as $x_0 = \frac{(1-p)q}{p(1-q)}$. Note that $x_0 < 1$ since $p > q$. We claim that $u \leq x_0$. To see why, we first note that $f'(x)$ can be rewritten as

$$x^{-q-1}(xp(1 - q) - (1 - p)q).$$

It is clear that $f'(x_0) = 0$. Since $xp(1 - q) - (1 - p)q$ is increasing in $x$, this further implies that $f'(x) > 0$ for $x > x_0$. Now, if $u > x_0$, then since $f'(x) > 0$ for $x > x_0$, we would conclude that $f(u) < f(1)$, which is not possible since $f(u) = f(1) = 1$. Therefore, $u \leq x_0$ as claimed.

We now claim that $x_0^{1-q} \leq 1 - p + q$. This would finish the proof since, together with $u \leq x_0$, this would imply

$$p(1 - u^{1-q}) \geq p(1 - x_0^{1-q}) \geq p(p - q) = \Omega(p - q),$$

where for the last equation we have used the assumption $p \in (c, 1 - c)$.

To prove the claim, define $\varepsilon := p - q$. We need to show that $x_0^{1-q} \leq 1 - \varepsilon$, or equivalently $\ln(x_0) \leq \frac{\ln(1-\varepsilon)}{1-q}$. By defintion of $x_0$, this is equivalent to

$$\ln\left(\frac{(1 - p)(p - \varepsilon)}{(1 - p + \varepsilon)p}\right) \leq \frac{1}{1 - p + \varepsilon} \ln(1 - \varepsilon). \tag{C.11}$$

Fix $p$ and consider both hand sides as a function of $\varepsilon$. Putting $\varepsilon = 0$, both hands side coincide as they both equal 0. To prove Euqation (C.11), it suffices to show that as we increase $\varepsilon$, the left hand side decreases faster than the right hand side. Equivalently, we need to show that the derivative of the LHS with respect to $\varepsilon$ is larger than the derivative of the RHS with respect to $\varepsilon$ for $\varepsilon \leq [0, p]$. Taking the derivative with respect to $\varepsilon$ on LHS, we obtain

$$\frac{d}{d\varepsilon}(\ln(1 - p) + \ln(p - \varepsilon) - \ln(1 - p + \varepsilon) - \ln(p)) = -\frac{1}{p - \varepsilon} - \frac{1}{1 - p + \varepsilon}.$$

Similarly taking the derivative on RHS we obtain

$$\frac{d}{d\varepsilon}\left(\frac{\ln(1 - \varepsilon)}{1 - p + \varepsilon}\right) = -\frac{1}{(1 - \varepsilon)(1 - p + \varepsilon)} - \frac{\ln(1 - \varepsilon)}{(1 - p + \varepsilon)^2}.$$

We therefore need to show that

$$\frac{-1}{1 - p + \varepsilon} + \frac{-1}{p - \varepsilon} \leq \frac{-1}{(1 - p + \varepsilon)(1 - \varepsilon)} + \frac{-\ln(1 - \varepsilon)}{(1 - p + \varepsilon)^2}. \tag{C.12}$$

We note however that

$$\frac{-1}{1 - p + \varepsilon} + \frac{-1}{p - \varepsilon} = \frac{\varepsilon - p - 1 + p - \varepsilon}{(1 - p + \varepsilon)(1 - \varepsilon)} = \frac{-1}{(1 - p + \varepsilon)(1 - \varepsilon)}.$$

Therefore Equation (C.12) is equivalent to

$$\frac{-\ln(1 - \varepsilon)}{(1 - p + \varepsilon)^2} \geq 0,$$

which is true since $\varepsilon \in [0, p]$. This proves the claim $x_0^{1-q} \leq 1 - \varepsilon$, finishing the proof. $\qquad \square$

We now prove Lemma C.2 using Lemma C.11 and C.12.

*proof of Lemma C.2.* Define the random variable $Y_i$ as $Y_i = X_{i+1} - q$. Note that $Y_i$ takes value $1 - q$ with probability $p$ and takes $-q$ with probability $1 - p$. Set $u$ to be the value specified in Lemma C.12. For $n \geq 0$, define $Z_n := u^{\sum_{i=1}^n Y_i}$. We first observe that $Z_n$ is a martingale with respect to $Y_1, \ldots, Y_n$ as

$$\mathbb{E}\left[Z_{n+1} | Y_1, \ldots Y_n\right] = \mathbb{E}\left[u^{\sum_{i=1}^{n+1} Y_i} | Y_1, \ldots Y_n\right] = u^{\sum_{i=1}^n Y_i} \cdot (p \cdot u^{1-q} + (1-p) \cdot u^{-q})$$

$$= u^{\sum_{i=1}^n Y_i} = Z_n.$$

Since $0 < u < 1$, this further implies

$$\Pr\left[\forall n \geq 0 : \sum_{i=1}^n Y_i \geq q - 1\right] = 1 - \Pr\left[\exists n \geq 0 : \sum_{i=1}^n Y_i < q - 1\right]$$

$$= 1 - \Pr\left[\max_{j \in [n]}\{u^{\sum_{i=1}^j Y_i}\} \geq u^{q-1}\right]$$

$$\geq 1 - \frac{\mathbb{E}[Z_1]}{u^{q-1}}$$

$$= 1 - u^{1-q},$$

where the first inequality follows from Lemma C.11 and the final equality follows from $\mathbb{E}[Z_1] = \mathbb{E}[Z_0] = \mathbb{E}[u^0] = 1$.

Since $Y_i$ is a function of $X_{s+1}$, we independently have $X_1 = 1$ with probability $p$. Therefore, with probability $p(1 - u^{1-q})$.

$$X_i = 1 \text{ and } \forall n \geq 1 : \sum_{i=2}^n (X_i - q) \geq q - 1,$$

which further implies $\sum_{i=1}^n (X_i - q) \geq 0$. Therefore,

$$\Pr\left[\forall n \geq 1 : \frac{\sum_{i=1}^n X_i}{n} \geq q\right] \geq p(1 - u^{1-q}) \geq \Omega(p - q),$$

where the inequality follows from Equation (C.10). $\qquad\square$

# D   Proofs from Section 4: Upper Bounds for Optimistic Agents

## D.1   Proof of Theorem 4.1 and Theorem 4.4

We define certain "clean events" to capture desirable realizations of random rewards, and decompose our regret bounds based on whether or not these events hold. The "clean events" ensure that the index of each arm is not too far from its true mean reward; more specifically, that the index is "large enough" for the good arm, and "small enough" for the bad arm. We have two "clean events", one for each arm, defined in terms of the reward-table as follows:

$$\texttt{Clean}_1^\eta := \left\{ \forall i \in [T] : \texttt{UCB}_{1,i}^{\texttt{tape}, \eta} \geq \mu_1 - \Delta/2 \right\}, \tag{D.1}$$

$$\texttt{Clean}_2^\eta := \left\{ \forall i \geq 64\, \eta/\Delta^2 : \texttt{UCB}_{2,i}^{\texttt{tape}, \eta} \leq \mu_2 + \Delta/4 \right\}. \tag{D.2}$$

Our analysis is more involved compared to the standard analysis of the UCB1 algorithm [6], essentially because we cannot make $\eta$ be "as large as needed" to ensure that clean events hold with very high probability. For example, we cannot upper-bound the deviation probability separately for each round and naively take a union bound over all rounds.[20] Instead, we apply a more careful "peeling technique", used *e.g.,* in Audibert and Bubeck [5], so as to avoid *any* dependence on $T$ in the lemma below.

---

[20]Indeed, this would only guarantee that clean events hold with probability at least $1 - O(T \cdot e^{-\Omega(\eta)})$, which in turn would lead to a regret bound like $O(T^2 \cdot e^{-\Omega(\eta)})$.

**Lemma D.1.** *The clean events hold with probability*

$$\Pr\left[\,\texttt{Clean}_1^{\eta}\,\right] \geq 1 - O\left(\,(1 + \log(1/\Delta)) \cdot e^{-\Omega(\eta)}\,\right), \tag{D.3}$$

$$\Pr\left[\,\texttt{Clean}_2^{\eta}\,\right] \geq 1 - O\left(\,e^{-\Omega(\eta)}\,\right). \tag{D.4}$$

We show that under the appropriate clean events, $\eta$-optimistic agents cannot play the bad arm too often. In fact, this claim extends to $[\eta, \eta_{\max}]$-optimistic agents.

**Claim D.2.** *Assume that events $\texttt{Clean}_1^{\eta}$ and $\texttt{Clean}_2^{\eta_{\max}}$ hold. Then $[\eta, \eta_{\max}]$-optimistic agents cannot choose the bad arm more than $64\,\eta_{\max}/\Delta^2$ times.*

*Proof.* For the sake of contradiction, suppose $[\eta, \eta_{\max}]$-optimistic agents choose the bad arms at least $n = 64\,\eta_{\max}/\Delta^2$ times, and let $t$ be the round when this happens. However, by event $\texttt{Clean}_1^{\eta}$, the index of arm 1 is at least $\mu_1 - \Delta/2$. By event $\texttt{Clean}_2^{\eta_{\max}}$, the index of arm 2 is at least $\texttt{UCB}_{i,n}^{\texttt{tape}, \eta} \leq \mu_2 + \Delta/4$, which is less than the index of arm 1, contradiction. $\qquad\square$

For the "joint" clean event, $\texttt{Clean} := \texttt{Clean}_1^{\eta} \cap \texttt{Clean}_2^{\eta_{\max}}$, Lemma D.1 implies

$$\Pr\left[\,\texttt{Clean}\,\right] \geq 1 - O\left(\,\log\left(1/\Delta\right) \cdot e^{-\Omega(\eta)}\,\right). \tag{D.5}$$

When the clean events fail, we upper-bound regret by $\Delta \cdot T$, which is the largest possible. Thus, Lemma D.2 and Eq. (D.5) imply Theorem 4.4, which in turn implies Theorem 4.1 as a special case.

## D.2   Proof of Theorem 4.5

We reuse the machinery from Section D.1, but we need some extra work. Recall that all agents are assumed to be $\eta_{\max}$-confident, whereas only a fraction are optimistic. Essentially, we rely on the optimistic agents to sample the good arm sufficiently many times (via Claim D.2). Once this happens, all other agents "fall in line" and cannot choose the bad arm too many times.

In what follows, let $m = 1 + 64\,\eta_{\max}/\Delta^2$.

**Claim D.3.** *Assume $\texttt{Clean}$. Suppose the good arm is sampled at least $m$ times by some round $t_0$. Then after round $t_0$, agents cannot choose the bad arm more than $m$ times.*

*Proof.* For the sake of contradiction, suppose agent $t \geq t_0$ has at least $m$ samples of the bad arm (*i.e.*, $n_{2,t} \geq m$), and chooses the bad arm once more. Then the index of the good arm satisfies

$$\begin{aligned}
\texttt{Ind}_{1,t} &\geq \texttt{LCB}_{1,t}^{\eta_{\max}} && \textit{($\eta_{\max}$-confident agents)}\\
&\geq \texttt{LCB}_{1,m}^{\texttt{tape}, \eta_{\max}} && \textit{(by definition of $t_0$)}\\
&\geq \texttt{UCB}_{1,m}^{\texttt{tape}, \eta_{\max}} - 2\sqrt{\eta_{\max}/m} && \textit{(by definition of UCBs/LCBs)}\\
&\geq \texttt{UCB}_{1,m}^{\texttt{tape}, \eta} - 2\sqrt{\eta_{\max}/m} && \textit{(since $\eta_{\max} \geq \eta$)}\\
&> \mu_1 - \Delta/2 && \textit{(by $\texttt{Clean}_1^{\eta}$ and the definition of $m$).}
\end{aligned}$$

The index of the bad arm satisfies

$$\begin{aligned}
\texttt{Ind}_{2,t} &\leq \texttt{UCB}_{1,t}^{\eta} && \textit{($\eta$-confident agents)}\\
&\leq \mu_2 + \Delta/4 && \textit{(by $\texttt{Clean}_1^{\eta}$ and the definition of $m$),}
\end{aligned}$$

which is strictly smaller than $\texttt{Ind}_{1,t}$, contradiction. $\qquad\square$

For Claim D.3 to "kick in", we need sufficiently many optimistic agents to arrive by time $t_0$. Formally, let $\mathcal{E}_t$ be the event that at least $2m$ agents are $[\eta, \eta_{\max}]$-optimistic in the first $t$ rounds.

**Corollary D.4.** *Assume $\texttt{Clean}$. Further, assume event $\mathcal{E}_{t_0}$ for some round $t_0$. Then (by Claim D.2) the good arm is sampled at least $m$ times before round $t_0$. Consequently (by Claim D.3), agents cannot choose the bad arm more than $m + t_0$ times.*

Finally, it is easy to see by Chernoff Bounds that $\Pr\left[\,\mathcal{E}_{t_0}\,\right] \geq 1 - e^{-\Omega(\eta)}$ for some $t_0 = O(m/q)$, where $q$ is the probability from the theorem statement. So, $\Pr\left[\,\texttt{Clean} \cap \mathcal{E}_{t_0}\,\right]$ is lower-bounded as in Eq. (D.5). Again, when $\texttt{Clean} \cap \mathcal{E}_{t_0}$ fails, we upper-bound regret by $\Delta \cdot T$. So, Corollary D.4 and the lower bound on $\Pr\left[\,\texttt{Clean} \cap \mathcal{E}_{t_0}\,\right]$ implies the theorem.

## D.3 Proof of Lemma D.1

We assume without loss of generality that $\eta > 2$. If $\eta \leq 2$, the Lemma's statement can be made vacuous using large enough constants in $O$. In addition, for mathematical convenience, we will assume that the tape for each arm is infinite, even though the entries after $T$ will never actually be seen by any of the agents.

For each arm $a$, we first separately consider each interval of the form $[n, 2n]$ and bound the probability that $\text{UCB}_{a,i}^{\texttt{tape}, \eta}$ deviates too much from $\mu_a$ for $i \in [n, 2n]$. While this can be done crudely by applying a union bound over all $i$, we use the following maximal inequality.

**Lemma D.5** (Eq. (2.17) in [32]). *Given a sequence of i.i.d. random variables $(X_i)_{i \in [n]}$ in $[0, 1]$ such that $\mathbb{E}[X_i] = \mu$, the inequality states that for any $x > 0$,*

$$\Pr\left[\exists i \in [n] : \left|\sum_{j=1}^{i} (X_j - \mu)\right| > x\right] \leq 2e^{-\frac{2x^2}{n}}.$$

Focusing on some interval of the form $[n, 2n]$ for $n \in \mathbb{N}$, and applying this inequality to the reward tape of arm $a$, we conclude that

$$\Pr\left[\exists i \in [n, 2n] : \left|\widehat{\mu}_{a,i}^{\texttt{tape}} - \mu_a\right| \geq x\right] \leq O(e^{-\Omega(nx^2)}). \tag{D.6}$$

Define $f := \lceil 64\eta/\Delta^2 \rceil$. We note that $f = \Theta(\eta/\Delta^2)$ given the assumption $\eta > 2$. In order to bound $\Pr[\texttt{Clean}_2^\eta]$, we will apply this inequality to each interval $[n, 2n]$ for $n \geq f$, and take a union bound. Formally,

$$
\begin{aligned}
1 - \Pr[\texttt{Clean}_2^\eta] &\leq \Pr\left[\exists i \geq f : \widehat{\mu}_{2,i}^{\texttt{tape}} > \mu_2 + \Delta/8\right] && \textit{(Since } \sqrt{\eta/i} \leq \Delta/8 \textit{ for } i \geq f\textit{)} \\
&\leq \sum_{r=0}^{\infty} \Pr\left[\exists i \in [f2^r, f2^{r+1}] : \widehat{\mu}_{2,i}^{\texttt{tape}} > \mu_2 + \Delta/8\right] && \textit{(Union bound)} \\
&\leq O\left(\sum_{r=0}^{\infty} e^{-\Omega(\eta 2^r)}\right) && \textit{(By Eq. (D.6))} \\
&\leq O\left(\sum_{r=0}^{\infty} e^{-\Omega(\eta(r+1))}\right) && \textit{(Since } 2^r \geq r+1 \textit{ for } r \in \mathbb{N}\textit{)} \\
&= O\left(\frac{1}{e^{\Omega(\eta)} - 1}\right) && \textit{(Sum of geometric series)} \\
&\leq O(e^{-\Omega(\eta)}) && \textit{(By } \eta > 2\textit{)}
\end{aligned}
$$

In order to bound $\Pr[\texttt{Clean}_1^\eta]$, we separately handle the intervals $n < f$ and $n \geq f$. For $n \geq f$, repeating the same argument as above for arm 1 implies

$$\Pr\left[\exists i \geq f : \widehat{\mu}_{1,i}^{\texttt{tape}} < \mu_1 - \Delta/8\right] \leq O(e^{-\Omega(\eta)}).$$

For $n < f$, we use a modified argument that utilizes the extra $\sqrt{\eta/i}$ term in $\text{UCB}_{1,i}^{\texttt{tape}, \eta}$. Instead of bounding the probability $\widehat{\mu}_{1,i}^{\texttt{tape}}$ having deviation $\Delta/8$, we bound the probability that it deviates by

$\sqrt{\eta/i}$. This results in a marked improvement because $\sqrt{\eta/i}$ increases as we decrease $i$. Formally,

$$\Pr\left[\exists i \in [1, f] : \widehat{\mu}_{1,i}^{\texttt{tape}} < \mu_1 - \sqrt{\eta/i}\right]$$

$$\leq \sum_{r=0}^{\lceil \log(f) \rceil} \Pr\left[\exists i \in [2^r, 2^{r+1}] : \widehat{\mu}_{1,i}^{\texttt{tape}} < \mu_1 - \sqrt{\eta/i}\right] \qquad \textit{(Union bound)}$$

$$\leq \sum_{r=0}^{\lceil \log(f) \rceil} \Pr\left[\exists i \in [2^r, 2^{r+1}] : \widehat{\mu}_{1,i}^{\texttt{tape}} < \mu_1 - \sqrt{\eta/2^{r+1}}\right] \qquad \textit{(By assumption on $i$)}$$

$$\leq O\left(\sum_{r=0}^{\lceil \log(f) \rceil} e^{-\Omega(\eta)}\right) \qquad \textit{(By Eq. (D.6))}$$

$$= O(\lceil \log(f) \rceil e^{-\Omega(\eta)}).$$

Finally, we note that since $\eta > 2$,

$$\lceil \log(f) \rceil \leq O(1 + \log(f)) = O(1 + \log(\eta) + \log(1/\Delta)).$$

This implies Eq. (D.3) because $O(\log(\eta)e^{-\Omega(\eta)})$ can be rewritten as $O(e^{-\Omega(\eta)})$ by changing the constant behind $\Omega$.

## E Proofs from Section 5: Learning Failures for Bayesian Agents

In this section, we prove Theorem 5.1. We first briefly review some properties of the beta distribution. Throughout the section, we consider a beta distribution with parameters $\alpha, \beta$.

**Lemma E.1** (Fact 1 in [3]). *Let $F_{n,p}^B$ denote the CDF of the binomial distribution with paramters $n, p$ and $F_{\alpha,\beta}^{beta}$ denote the CDF of the beta distribution. Then,*

$$F_{\alpha,\beta}^{beta}(y) = 1 - F_{\alpha+\beta-1,y}^B(\alpha - 1)$$

*for $\alpha, \beta$ that are positive integers.*

Using Hoeffding's inequality for concentration of the binomial distribution, we immediately obtain the following corollary.

**Corollary E.2.** *Define $\rho_{\alpha,\beta} := \frac{\alpha-1}{\alpha+\beta-1}$. If $X$ is sampled from the beta distribution with parameters $(\alpha, \beta)$,*

$$\Pr\left[\,|X - \rho_{\alpha,\beta}| \leq y\,\right] \leq 2e^{-(\alpha+\beta-1)y^2}.$$

*In addition, letting $Q(.)$ denote the quantile function of the distribution,*

$$[Q(\zeta), Q(1 - \zeta)] \subseteq \left[\rho_{\alpha,\beta} - \sqrt{\frac{\ln(2/\zeta)}{\alpha + \beta - 1}}, \rho_{\alpha,\beta} + \sqrt{\frac{\ln(2/\zeta)}{\alpha + \beta - 1}}\right],$$

Let $\alpha_{a,n}, \beta_{a,n}$ denote the posterior distribution after observing $n$ entries of the tape for arm $a$. Note that since we are assuming independent priors, the posterior for each arm is independent of the seen rewards of the other arm. Define $M_{a,n} := \alpha_{a,n} + \beta_{a,n}$. We note that by definition, $\alpha_{a,0}, \beta_{a,0}$ coincide with the prior $\alpha_a, \beta_a$. We analogously define $M_a := \alpha_a + \beta_a$. Define $\rho_{a,n} := \frac{\alpha_{a,n}-1}{M_{a,n}-1}$ and $\xi_{a,n} := \frac{\alpha_{a,n}}{M_{a,n}}$. We note that $\xi_{a,n}$ is the mean of the posterior distribution after observing $n$ entries of arm $a$.

**Lemma E.3.** *For all $n \geq 0$, $\left|\widehat{\mu}_{a,n}^{\texttt{tape}} - \xi_{a,n}\right| \leq O\left(\frac{M_{a,0}}{n+M_{a,0}}\right)$.*

*Proof.* After observing $n$ entries, the posterior parameters satisfy

$$\alpha_{a,n} := \alpha_{a,0} + \sum_{i \leq n} \texttt{Tape}_{a,i}, \quad \beta_{a,n} := \beta_{a,0} + \sum_{i \leq n}(1 - \texttt{Tape}_{a,i}).$$

It follows that

$$\xi_{a,n} = \frac{\alpha_{a,0} + \sum_{i \leq n} \texttt{Tape}_{a,i}}{\alpha_{a,0} + \beta_{a,0} + n}.$$

Defining $X := \sum_{i \leq n} \texttt{Tape}_{a,i}$, we can bound the difference between $\xi_{a,n}$ and $\widehat{\mu}_{a,n}^{\texttt{tape}}$ as

$$
\begin{aligned}
\left| \frac{\alpha_{a,0} + X}{M_{a,0} + n} - \frac{X}{n} \right| &= \left| \frac{n\alpha_{a,0} + nX - nX - XM_{a,0}}{n(n + M_{a,0})} \right| \\
&= \left| \frac{n\alpha_{a,0} - XM_{a,0}}{n(n + M_{a,0})} \right| \\
&\leq \frac{\alpha_{a,0}}{n + M_{a,0}} + \frac{M_{a,0}}{n + M_{a,0}} \qquad \text{(Since } X \leq n) \\
&\leq O\left( \frac{M_{a,0}}{n + M_{a,0}} \right)
\end{aligned}
$$

$\square$

**Lemma E.4.** *For all* $n \geq 0$, $|\xi_{a,n} - \rho_{a,n}| \leq O\left( \frac{1}{n + M_{a,0}} \right)$.

*Proof.*

$$
\begin{aligned}
\left| \frac{\alpha_{a,n} - 1}{M_{a,n} - 1} - \frac{\alpha_{a,n}}{M_{a,n}} \right| &= \left| \frac{-M_{a,n} + \alpha_{a,n}}{M_{a,n}(M_{a,n} - 1)} \right| \\
&\leq \frac{M_{a,n}}{M_{a,n}(M_{a,n} - 1)} \\
&= \frac{1}{M_{a,n} - 1} \\
&= O\left( \frac{1}{n + M_{a,0}} \right) \qquad \text{(Since } M_{a,n} = M_{a,0} + n \text{ and } M_{a,0} \geq 1)
\end{aligned}
$$

$\square$

We can now prove Theorem 5.1.

*Proof of Theorem 5.1.* We start with part (a). Set $\eta$ to be large enough such that

$$\left| \widehat{\mu}_{a,n}^{\texttt{tape}} - \xi_{a,n} \right| \leq \sqrt{\frac{\eta}{n}}.$$

Since $\frac{M_a}{n + M_a} \leq \frac{M_a}{n}$, by Lemma E.3, this can be achieved with $\eta \geq O(M_a / \sqrt{N_0})$, which proves part (a).

For part (b), set $\eta$ to be large enough such that $\left| \widehat{\mu}_{a,n}^{\texttt{tape}} - \rho_{a,n} \right| \leq \frac{1}{2} \cdot \sqrt{\frac{\eta}{n}}$. Given, Lemmas E.3 and E.4, this can be achieved with $\eta \geq O(M_a / \sqrt{N_0})$. Since $M - 1 \geq n$, we can further gaurantee $\frac{\ln(2/\zeta)}{M-1} \leq \frac{\eta}{4n}$ by setting $\eta \geq O(\ln(1/\zeta))$, which finishes the proof together with Corollary E.2. $\square$

# F  The corollary from Section 6

As a corollary of Theorem 6.1, we derive a 0-sampling failure, leading to $\Omega(T)$ Bayesian regret. Specifically, the agents never try arm 2 *when it is in fact the best arm*. This happens whenever the prior is independent across arms and has a positive density on the entire $[0, 1]$ interval.

**Corollary F.1.** *In the setting of Theorem 6.1, suppose the prior $\mathcal{P}$ is independent across arms and has a positive density for each arm (i.e., has probability density function that is strictly positive on $[0, 1]$). Then $\mathbb{E}[\mathrm{Regret}(T)] \geq c_{\mathcal{P}} \cdot T$, where the constant $c_{\mathcal{P}} > 0$ depends only on the prior $\mathcal{P}$.*

This follows from a more explicit, but more cumbersome corollary.

**Corollary F.2.** *Denote $\mu_1^0 = \mathbb{E}[\mu_1]$ and $\mu_2^0 = \mathbb{E}[\mu_2]$. Consider independent priors such that $\Pr[\mu_1 = 1] < (\mu_1^0 - \mu_2^0)/2$. Pick any $\alpha > 0$ such that $\Pr[\mu_1 \geq 1 - 2\alpha] \leq (\mu_1^0 - \mu_2^0)/2$. Then Bayesian regret is at least $T \cdot \left(\alpha/2\,(\mu_1^0 - \mu_2^0)\Pr[\mu_2 > 1 - \alpha]\right)$.*

*Proof.* Let $\mathcal{E}_1$ be the event that $\mu_1 < 1 - 2\alpha$ and arm 2 is never chosen. By Theorem 6.1 and the definition of $\alpha$, we have $\Pr[\mathcal{E}_1] \geq (\mu_1^0 - \mu_2^0)/2$.

Let $\mathcal{E}_2$ be the event that $\mu_2 > 1 - \alpha$. Under event $\mathcal{E}_1 \cap \mathcal{E}_2$, each round contributes $\mu_2 - \mu_1 \geq \alpha$ to regret, so $\mathbb{E}[R(T) \mid \mathcal{E}_1 \cap \mathcal{E}_2] \geq \alpha T$.

Since event $\mathcal{E}_1$ is determined by the prior on arm 1 and the rewards of arm 2, it is independent from $\mathcal{E}_2$. It follows that

$$\mathbb{E}[R(T)] \geq \mathbb{E}[R(T) \mid \mathcal{E}_1 \cap \mathcal{E}_2] \cdot \Pr[\mathcal{E}_1 \cap \mathcal{E}_2]$$
$$\geq \alpha T \cdot (\mu_1^0 - \mu_2^0)/2 \cdot \Pr[\mathcal{E}_2]. \qquad \square$$

