# OpenReview forum: "Bandit Social Learning under Myopic Behavior"
_NeurIPS.cc/2023/Conference — NeurIPS 2023 poster_

### Official Review · Reviewer_Nim8 · 2023-06-20

**Soundness:** 3 good
**Presentation:** 2 fair
**Contribution:** 2 fair
**Rating:** 6
**Confidence:** 4

**Summary:**

The paper considers social learning in a two-armed Bernoulli bandit scenario, where agents sequentially arrive and pull an arm with the highest index, where the index is arbitrarily chosen to be within some confidence bound of the empirical mean of the arm, parametrized by $\eta$. This behavior subsumes greedy behavior ($\eta = 0$) and regret-optimal policies such as UCB1. The main contribution is a tight characterization of the probability of learning failure, i.e., most agents will pull the suboptimal arm as a function of the reward gap and $\eta$. They extend this result to the case where agents are Bayesian and use truncated priors to inform indices.

**Strengths:**

- As the paper suggests, while the fact that greedy behavior causes learning failure with constant probability in standard classes of K armed bandit problems is folklore, there is no formal study of the boundaries of regimes where greedy behavior starts failing. Thus the motivation of the paper is solid.
- The techniques for proving the lower bounds on the probabilities of failure are novel relative to standard bandit literature.

**Weaknesses:**

- The paper's conceptual takeaways are not surprising, and the technical contribution adds little beyond tightly characterizing the probabilities of failure as a function of $\eta$, the value of which is unclear for such a specific model of social learning.
- The examples where the greedy behavior suffices in prior literature are contextual bandit environments where context diversity is the driver of exploration. So it seems that understanding the boundaries of greedy behavior must work with some interpolation between contextual and independent armed environments. Instead, the present paper focuses on a two-armed bandit environment where the confidence bounds are parametrized, the value of which is unclear since one anticipates learning failure for any fixed parameter value.
- The interpretation of the results for the non-Bayesian setting is obfuscated by the dependency on $N_0$, the initial number of samples. It seems that a lot of technical maneuvering (e.g., assumption 3.2) arises because, when $N_0$ is small, one cannot eliminate the possibility of avoiding learning failure due to the confidence bounds being truncated at the boundary of $[0,1]$. This seems orthogonal to the central issue of focus (in this regard, Theorem 3.9 certainly appears to be cleaner). This makes the results appear too technical without adding anything substantial to the dialogue on the sufficiency of greedy algorithms for bandit learning.


**Questions:**

Could you comment on the qualitative difference between Theorem 3.9 and Theorem 3.1, and whether focusing on Theorem 3.9 with the very reasonable assumptions in P1-P2 would achieve the goals of the paper?

---------------------------
Post rebuttal: Thanks for the clarifications. While my original concerns about the practicality of regimes and the surprise-factor remain, I see the conceptual value of the results and believe they deserve to be published. I have raised my score accordingly.


**Limitations:**

The results pertain to a very specific two-armed bandit model and so the paper is explicit about its limitations and applicability.

---

> ### Author Rebuttal · Authors · 2023-08-09
>
> Thanks for the thoughtful review which raises several rather subtle issues. Let us address the stated weaknesses point-by-point.
>
> **[W1]** What might count as “surprising” is that we managed to *prove* any/all these things, let alone in such generality. Given that such results remained unproved for so long, despite its obvious foundational value, we believe that success was very unclear in hindsight. (Note that proving “expected” results is often valuable – as a super-extreme example, consider proving P\neq NP.)
>
> One “conceptual takeaway” that may actually be surprising per se is that we provide a lower bound on the failure probability that does not degrade with the “strength of initial beliefs”. This point is most clearly stated in the “fully Bayesian” result (Thm 6.1), where the failure probability is lower-bounded by the Bayesian-expected gap, independently of the Bayesian prior. For the frequentist results in Sec3, the situation is a bit more subtle: “strength of beliefs” is driven by $N_0$, and the non-trivial regime is $\Delta < O(1/\sqrt{N_0})$ (otherwise the initial data alone essentially resolves the best arm). Then we get a lower bound of $\Delta \cdot e^{O_c(\eta)}$. This should be contrasted with the “naive” failure mode when all examples from the good arm fail, which only happens with probability exponential in $N_0$ (see Remark 3.7).
>
>
> For good measure, let us recap our other "conceptual takeaways":
> - Greedy algorithm fails, essentially for any two-armed bandit instance.
> - In fact, any myopic behavior fails, with "severity" driven by $\eta$. (More on this under "generality" in the general rebuttal.)
> - Optimism is essentially optimal for a given $\eta$, while pessimism does much worse.
> - Small fraction of optimists goes a long way.
> - The failure results admit both "frequentist" and "Bayesian" framing.
>
> We believe the significance of $\eta$ is very clear: it defines the range of permissible behaviors, has a clear interpretation in terms of confidence intervals, and drives the “severity” of failures throughout various twists and turns of our technical story.
>
> **[W2]** We completely agree that a good path forward is to interpolate between a few complex environments where the greedy algorithm is known to work and the simple structure(s) in which it is known to fail. In fact, this direction is/was very much on our radar!  However, we believe our current paper is a necessary precursor – and a badly missing one! Please also see para1 and “greedy” in the general rebuttal.
>
> A minor point: what drives the positive results on Greedy in prior work is not just the diversity of contexts but also some structural assumption that enables aggregation (linearity in some papers, separability in some others).  A few other positive results are driven by a (very) large number of arms (under some additional assumptions).
>
>
> **[W3 and Q]** The meaning of $N_0$ is that it controls the “strength of initial beliefs”, in the a natural frequentist interpretation thereof as the amount of initial data. So, in a sense, it is more than just an annoying technicality that we need to account for.
>
> In a narrower technical sense, we need to consider larger $N_0$ in order to avoid a trivial failure mode when *all* initial samples of the good arm have reward 0 (see Remark 3.7). Then, indeed, we indeed need some “technical maneuvering” around the case of very small $N_0$. To answer your direct question, please see “not merging Thm 3.1 and Thm 3.9” in the general rebuttal.
>
> We emphasize, however, that there’s much more to our technical story than the distinction between Thm 3.1 and Thm 3.9, e.g., see “technical story” in the general rebuttal.

---

### Official Review · Reviewer_1URz · 2023-06-30

**Soundness:** 3 good
**Presentation:** 4 excellent
**Contribution:** 3 good
**Rating:** 7
**Confidence:** 3

**Summary:**

The paper posits a bandit social learning (BSL) problem, which consists of a multi-armed bandit (MAB) problem where at each round an arm is pulled by a newly arrived agent, as a function of the history. This is motivated by reviews on online platforms, where agents pick decisions sequentially based on past reviews. Compared to standard MAB, where a centralized algorithm is run to minimize regret, in BSL each agent acts myopically and can be, e.g., greedy, optimistic, or pessimistic w.r.t. confidence intervals constructed around the reward estimate for each arm. The authors analyze the 2-arms setup and provide several learning failure results in identifying the optimal arm. Notably, the learning "fails" when agents are greedy or pessimistic, while it achieves optimal regret when a small fraction of the agents are optimistic. This was a general belief in standard MAB, but to the best of authors' knowledge their results are the first ones to assess it theoretically. Similar learning failures are also established for Bayesian agents who act according to their posterior.

**Strengths:**

- The paper reads really well and the results are sound. Moreover, the authors did a good job of positioning it into the existing related literature.
- The introduced BSL problem is simple, albeit very interesting, and of practical relevance, e.g., in review systems.
- Although the flavor of the results is quite specific to the BSL problem (where agents have different myopic behavior), the negative results apply to the more general MAB, constituting a relevant contribution to the broader bandits community.


**Weaknesses:**

- The authors consider 2-armed bandits for the sake of their analysis and negative results. However, it is not clear how the picture would change in the presence of more arms.
- No experiments are performed. Although the paper is of theoretical nature, would be nice to demonstrate the proven failure probabilities and how the injected optimism facilitates learning.

**Questions:**

I would like to know the author's view on extending such results to more than 2 arms. In particular, is a bigger set of arms always more detrimental in terms of learning/exploration? Should the initial number of samples $N_0$ intuitively scale with the number of arms?

**Limitations:**

Limitations are adequately addressed.

---

> ### Author Rebuttal · Authors · 2023-08-09
>
> Thanks for the thoughtful and a largely positive review!
>
> **[W1 and Q]** Please see “K>2 arms” in the general rebuttal. In particular, adding more arms could affect the failure probability positively, negatively, or not at all, depending on the problem instance.
>
> Re the semantics of $N_0$: indeed, it would make sense to have $N_0$ samples of each arm. (Note, however, that we think of $N_0$ as an exogenous parameter.)
>
> **[W2]** We’ve included some experiments as requested, please see in the general rebuttal.

---

> > ### Comment · Reviewer_1URz · 2023-08-21
> >
> > Thank you for the responses. I keep my score.

---

### Official Review · Reviewer_piEt · 2023-07-02

**Soundness:** 4 excellent
**Presentation:** 4 excellent
**Contribution:** 3 good
**Rating:** 7
**Confidence:** 3

**Summary:**

This paper studies bandit social learning problem with two arms. Instead of aiming to design an efficient algorithm with theoretial guarantee, the authors demonstrate negative results regarding myoptic behaviors of agents. The main contribution of this paper is proving the regret lower bounds of $\eta$-confidence agents, together with nearly matching upper bounds, which explains why greedy algorithms i.e., always exploit, are not efficient and why UCB1 algorithm requires extreme optimism.

**Strengths:**

* This paper is well written and easy to follow. The author does a great job of explaining complex concepts in a clear and concise way.
* The proofs appear solid and complete to me that both of UCB and Bayesian agents are taken into consideration in this paper.

**Weaknesses:**

* I believe this is a good paper because it provides solid theoretical insights into why agents perform less optimistically as UCB-type algorithms perform worse, and how this degradation varies with the degree of optimism. However, I feel that the proofs do not introduce any new techniques, which lowers my overall score.

**Questions:**

It might be better to add experiment simulations to validate the theory.

*Typo*

Line 39: the number of agents $T$...: should here be number of time steps?

---

> ### Author Rebuttal · Authors · 2023-08-09
>
> Thanks for the thoughtful and a largely positive review!
>
> **[W]** Re techniques, please see “techniques” in the general rebuttal. We would also like to re-emphasize the generality of allowed behaviors, please see “generality” in the general rebuttal.
>
> **[Q]** We’ve included some experiments as requested, please see in the general rebuttal.
>
> Re Line 39, “the number of agents $T$”: in our setup a new agent shows up in each round, so agents and rounds are essentially the same.

---

> > ### Comment · Reviewer_piEt · 2023-08-20
> > **After rebuttal**
> >
> > Thanks authors for their detailed response. I think my concerns are resolved and I will keep my initial score. Good luck!

---

### Official Review · Reviewer_ZFQ2 · 2023-07-08

**Soundness:** 3 good
**Presentation:** 3 good
**Contribution:** 2 fair
**Rating:** 3
**Confidence:** 2

**Summary:**

The paper proposes the model of social learning under myopic behavior, where a 2-armed bandit problem is considered with agents that behave myopically. Upper and lower bounds on the probability that all but \leq n agents choose the bad arm are derived.

**Strengths:**

- The paper is well written and easy to follow.
- The considered model is novel.
- The bounds on failure probability are tight.

**Weaknesses:**

- The results are only limited to the case of 2 arms.
- Assumption 3.2 is strong and seems to be unnecessary. For example if mu_1 is very small, then a lot of initial samples N0 are required even if mu_2 is close to 1 which is an easy to solve case that does not require a lot of samples.
- The considered myopic strategies are limited to a few number of strategies (confident, unbiased, optimistic, pessimistic, bayesian).
- The results are not surprising and can be obtained using standard concentration inequalities in the bandit literature, e.g., see [1].
- As the paper is only concerned about failure probabilities without the need to decide on a strategy, I am not convinced about the importance of the results.

[1] Lattimore, Tor, and Csaba Szepesvári. Bandit algorithms. Cambridge University Press, 2020.

**Questions:**

Please see weaknesses.

**Limitations:**

Yes.

---

> ### Author Rebuttal · Authors · 2023-08-09
>
> Thanks for the thoughtful and explicit review. Let us respond point-by-point to the stated weaknesses, in the same order.
>
> **[W1]** Please see “K>2 arms” in the general rebuttal.
>
>
> **[W2]** Assumption (3.2) is not that strong in the theoretical sense: it merely requires $N_0$ to be larger than a constant. Essentially, it ensures that the confidence intervals are proper subintervals of [0,1] – which matters even if $\mu_1$ is close to 0 and $\mu_2$ is close to 1.
>
> This assumption is needed for our analysis of Thm 3.1. However, we do remove it later on – which is the whole point of Thm 3.11 – but at the cost of minor-but-technical assumptions on the behaviors and some substantial complications in the analysis. We’d be thrilled to get rid of this assumption in any simpler way, suggestions welcome!
>
>
> **[W3]** On the contrary, $\eta$-confidence in Thm 3.1 (and Thm 3.9) allows for a range of behaviors, incl. unbiased and pessimism/optimism as special cases. Please see “generality” in the general rebuttal.
>
> **[W4]** Our negative results critically rely on anti-concentration and martingale tools, which are **not** standard in bandit lower bounds. In particular, it is really unclear how to get them while only using concentration tools, let alone standard ones such as Azuma-Hoeffding.
>
> Our positive results require *non-standard* tools to handle concentration, and (more importantly) a rather delicate way to define “clean events” and argue about them. Please also see “techniques” in the general rebuttal.
>
>
> **[W5]** There are thousands of papers on designing bandit algorithms, but ours is the first one on learning failures. Thus, we believe we fill an important gap in the literature.
>
> Further, learning failures are a very common theme in the vast literature on social learning, and usually are considered a main result (and often *the* main result). Learning failures arise in different technical settings and due to different reasons; we discuss this literature in Appendix A.

---

> > ### Author Response · Authors · 2023-08-18
> >
> > Dear reviewer, we were wondering whether our response has addressed your concerns. We'd be more than happy to provide any additional clarifications.
> >
> > Regarding the "surprise" factor, please also see our response to Rev Nim8 ("[W1]" therein).

---

### Official Review · Reviewer_XcKk · 2023-07-16

**Soundness:** 4 excellent
**Presentation:** 3 good
**Contribution:** 3 good
**Rating:** 6
**Confidence:** 4

**Summary:**

This paper considers a social learning problem motivated by reviews on online platforms, where (myopic) users make (purchase) decisions based on historical reviews and generate new reviews in an online fashion. It considers several different user behavioral types, such as the confidence based optimistic, pessimistic and neutral users. In addition, the users could have Bayesian belief on the mean reward and act according to the posterior update based on the history. The paper characterizes several cases where the learning failure occurs i.e., when all but a few agents choose the bad arm.


**Strengths:**

1. This paper introduces and analyzes an interesting setup of social learning. I can see how this setup may be applied to many real world applications, such as online recommendation systems (especially in presence of purchase decisions). I also expect many research projects to follow up on this fundamental social learning model.
2. The technical tools used in this paper are somewhat different from the standard bandit literature.
3. The possible results as the implications regarding the optimistic agents are very interesting.

**Weaknesses:**

1. The presentation and structure of this paper requires some improvements. The current version seems to have many results scattered around the paper, making it hard for us readers to switch the context from one section to another.  Section 3 is named as "learning failure", but it actually considers several different setups (theorem 3.1, 3.9, 3.10). Why not merge theorem 3.1 and 3.9? It might also be a good idea to use a table to summarize the results under different setups, e.g., when it fails, when it doesn't.
2. Overall, I find the problem setting interesting, but the technical results are not that surprising. For example, Theorem 3.1 requires a fixed \eta for every agent, which more or less violates its motivation of social learning --- the agents tend to have heterogeneous behavior types. I also wish the paper could connect its results and setups to real world applications (instead of imagining purely idealized scenarios).
3. The entire paper chooses to only focus on the two arms case. I expect the author to point out the exact technical challenges (or practical motivations) that prevent the analysis to be extended to the general cases.

**Questions:**

Please see the points listed up the weakness part. I really liked the problem setup, and I am willing to raise my score if the authors could convince me of the technical/conceptual significance of their existing results.

---

> ### Author Rebuttal · Authors · 2023-08-09
>
> Thanks for the thoughtful review. Let us respond point-by-point to the stated weaknesses.
>
> **[W1]** The structure of the paper is driven (and necessitated) by the somewhat intricate collection of results, see “technical story” in the general rebuttal. Given the commonality / connections both in story and in techniques, we believe it makes sense to group all negative results in one section and discuss them jointly; likewise, all positive results and all “Bayesian beliefs” results. In particular, negative and positive results match “globally” but not “point-by-point”: instead each “side” has its own “sub-story”.
>
> Re “not merging Thm 3.1 and Thm 3.9”: please see the general rebuttal.
> Adding a table is a great idea, we’ll do it!
>
>
> **[W2]** Please note that our negative results allow heterogeneous behaviors. In Thm 3.1, “$\eta$-confidence” with fixed $\eta$ defines not a particular behavior but a wide range of allowed behaviors, including heterogeneity, see “generality” in the general rebuttal. Likewise Thm 3.9 (under minor restrictions of symmetry and monotonicity). Even Thm 3.10, while focusing on pessimists, allows for varying levels of pessimism.
>
> With this generality, we plausibly capture what myopic real-world agents might do when faced with a simple learning problem. Moreover, the particular behaviors that we capture are well-documented in the literature on behavioral economics and/or cognitive psychology (e.g., optimism/pessimism, probability matching, recency bias).
>
> We note, however, that results for homogeneous behavioral types are quite common in the literature, as they tend to make the intended points in a most concrete and elegant way. For example, Thm 4.1 obtains a clean upper bound that matches our negative results. (Meanwhile, Thm 4.4 and Thm 4.5 drill deeper to investigate heterogeneity.)
>
>
> **[W3]** Please see “K>2 arms” in the general rebuttal.

---

> > ### Comment · Reviewer_XcKk · 2023-08-18
> >
> > I appreciate the authors' detailed response. After reading the rebuttal and other reviews, I decide to maintain my initial score.

---

> > > ### Author Response · Authors · 2023-08-18
> > >
> > > Thanks for your response! We would appreciate if you could point out whether we've addressed some of your concerns, and what are the remaining ones. Regarding what might count as "surprising", please also see our response to Rev Nim8 ("[W1]").
> > >
> > > Many thanks,
> > > The authors

---

### Author Rebuttal · Authors · 2023-08-09

Thanks for the thoughtful reviews. Many of our points are relevant to several reviews at once.

**[SIMULATIONS: NEW]** As requested, we provide simulations to illustrate our main findings. We focus on the fundamental regime when agents are homogeneously all $\eta$-optimistic (resp., all $\eta$-pessimistic) for some fixed $\eta\geq 0$. Mirroring our negative results, we investigate the probability of a learning failure. Consider the event $F_t$ that the bad arm is chosen in all rounds between $t$ and the time horizon $T$. We re-run the simulation 1000 times, and plot the fraction of runs for which $F_t$ happens, as a curve over time $t$.

We plot such curves for several representative values of $\eta$, ranging from LCB to greedy to UCB. Qualitatively, we find significant failures which (predictably) get worse as $\eta$ decreases (treating LCBs as negative $\eta$). We consider mean rewards $0.5 \pm \epsilon$, with  $\epsilon=0.05$ ("large gap", top) and $\epsilon=0.01$ ("small gap", middle).

We also investigate UCBs with larger $\eta$, and find similar failures (with smaller probs) at a larger $T$. (We check that including "weaker" failure modes would not change this plot by much.)

We'll be happy to include [even] more refined simulations in the final paper, if requested.
\
\
**[K>2 arms]** We emphasize that 2 arms is the fundamental case for negative results, which are the main theme in our paper. (The purpose of our positive results is to better characterize the failures.) Besides, many papers on social learning concentrate on two arms as a fundamental case, even for positive results. So, we strive to fully understand the 2-armed case, which – as we find -  allows for elegant guarantees, yet requires a rather complex “technical story” (see below).

That said, our negative results trivially extend to some instances with K>2 arms: e.g., just add K-2 arms with 0 reward. For some other instances, failure probabilities get much smaller (e.g., with many “best” arms, all of them need to experience a “bad” random event), or much larger (e.g., one good arm and many equally bad arms). We can spell out such extensions in the revision.

However, a general characterization of failure probability for K>2 arms is likely to be much more cumbersome, as it would now depend on several arms. While it may be within reach given our techniques, the “technical story” in such characterization is likely to be quite complex (e.g., as complex as the story in our paper, and possibly more so). We feel strongly that we are already near the limit of what one could put into a single paper, both conceptually and because of the page limit.
\
\
**[Generality]** While focusing on the basic learning problem, we allow considerable generality on the “behavioral side” (Lines 32-38, 183-189). We allow any behaviors consistent with the confidence intervals, possibly randomized and/or correlated across arms. In addition to greedy/unbiased behavior and varying levels of optimism/pessimism, this includes, e.g., versions of “active arms elimination” and “Thompson Sampling” (a.k.a. “probability matching” in behavioral economics). These behaviors can be arbitrarily different across agents, and also across arms (e.g., optimism on one arm, pessimism on the other). We also accommodate a form of “recency bias”, whereby one is more optimistic about an arm if more recent observations are better than the older ones. All these behaviors are well-documented in the literature on behavioral economics. We will include a more detailed discussion in the revision.
\
\
**[Greedy]** We believe the failure analysis of the greedy algorithm should stand on its own, as a badly missing foundational piece for bandit theory. While consistent with our expectations, it was not clear in hindsight which assumptions would be needed and what would be the “strength” and generality of the learning failures.
\
\
**[Techniques]**  Our lower-bound analysis relies on anti-concentration and martingale tools, which are not very standard per se (see Lines 280-285), and very non-standard for bandit lower bounds. However, the main technical complexity is in _applying_ these tools, i.e., setting up the right events and arguing what happens when these events hold (especially so for Thm 3.9). The positive results reuse the standard UCB machinery, but require a much more delicate setup and analysis of the “clean events” (particularly so in Thm 4.5). Unfortunately, we could only hint at all this complexity in the body of the paper! (Lines 85-103). Our techniques and tricks provide foundation for subsequent work on bandit social learning in more complex learning problems.
\
\
**[Technical story]** The technical story in our paper is quite intricate, illustrating the complexity of the problem even for 2 arms, and necessitating a particular order of presentation. The story  proceeds from the main failure result (Thm 3.1) to handling a somewhat trickier case of small $N_0$ (Thm 3.9), to proving a much stronger failure for pessimistic agents (Thm 3.10). Bayesian beliefs are handled as a special case (Thm 5.1). The upper bounds proceed from uniform optimism (Thm 4.1) to upwards-varying optimism (Thm 4.4) to a small fraction of optimists  (Thm 4.5). Finally, there’s a much stronger result in a “fully Bayesian” setup, with a different proof.
\
\
**[Not merging theorems Thm 3.1 and Thm 3.9]** Merging them would lose the appealing generality of behaviors in Thm 3.1, as well as its relative simplicity compared to Thm 3.9 (both in the statement and in the analysis). Further, the symmetry assumption (P1) in Thm 3.9 may actually break if an agent gives more “benefit of a doubt” to one of the arms when both arms look very bad. (E.g., what if when both Chinese and Italian restaurants look bad, one chooses Chinese.)  This is why we chose to separate these two results, specialized Thm 3.9 for the case of very small $N_0$, and in fact used Thm 3.1 to *motivate* Thm 3.9.

---

> ### Comment · Area_Chair_hYrz · 2023-08-17
>
> Dear reviewers,
>
> Thank you all for your time and dedication. We are approaching the end of author rebuttal phase. Could you please read the rebuttal and see if they addressed your questions or if there are remaining questions?
>
> Thanks,
>
> AC

---

### Decision · Program_Chairs · 2023-09-21

**Decision:**

Accept (poster)

**Comment:**

The paper studied bandit social learning problem motivated by reviews on online platforms, where myopic users make decisions based on historical reviews and generate new reviews on the fly. The paper considered two-armed Bernoulli bandits where the myopic behavior follows choosing the index of action within some confidence bound of the empirical mean of the arm parametrized by $\eta$. The authors demonstrate negative results regarding myoptic behaviors of agents. Understanding when failure occurs is a novel result and a major contribution of the paper. The authors provide the regret lower bounds of $\eta$-confidence agents, together with nearly matching upper bounds, which explains why greedy algorithms are not efficient and why UCB1 algorithm requires extreme optimism. Overall, the reviewers appreciate the novel problem setting, the clear writing, the sound theoretical results, and the insights behind the negative results. One reviewer has concerns regarding the limitation of two-armed bandit setting and strong assumptions, but the authors' response addressed those concerns in my opinion. I recommend acceptance.